# HYPERSYN: SYNTHESIZING INSTANCE-WISE MODEL BY FUSING BLACKBOX EXPERT VIA HYPERNETWORK

## ABSTRACT

Pretrained experts are now ubiquitous, encouraging their ensembling to achieve improved performance. However, in many scenarios, they are exposed only through prediction APIs, creating black-box settings where weights and internal representations are unavailable. Existing black-box ensembling methods often perform poorly under out-of-domain conditions, since they rely solely on expert outputs. To highlight this limitation, we identify three types of data regions and show how current methods fail in certain cases. To address these challenges, we propose HyperSyn, a deep learning framework that synthesizes an instance-specific model for each data point using expert outputs as input to a hypernetwork. HyperSyn naturally fits the black-box setting and provides greater expressiveness, particularly when existing experts fail on unseen test domains. Extensive experiments on both synthetic and real-world datasets demonstrate that HyperSyn outperforms commonly used ensemble techniques and achieves state-of-the-art performance when the data exhibit complex and unknown domain structure.

## 1 INTRODUCTION

Modern machine learning systems increasingly rely on pools of pretrained experts—such as foundation models and domain specialists on large datasets that are often unknown and unavailable due to privacy and or scale constraints. Jointly retraining or fine-tuning all experts on a combined corpus is rarely feasible, motivating post-hoc strategies that combine frozen experts at deployment. The pragmatic question we study goes beyond "mimic the pool": given only black-box access to frozen experts and a modest labeled reference set, can we build a new predictor that (i) achieves low absolute risk and (ii) is, on average, at least as good as the best frozen expert? We call the latter property oracle dominance, a safety criterion that prevents degradation relative to the expert pool.

Prior work spans classical ensembles (bagging/boosting/stacking), mixture-of-experts (MoE) with input-conditioned routing(Breiman, 1996; Freund & Schapire, 1996; Friedman, 2001; Ganaie et al., 2022), nonlinear fusion of expert outputs(Wang et al., 2024), and *white-box* parameter-space methods that merge weights or exchange internal representations(Hao Zhao, 2024; Tang et al., 2024). White-box approaches can be powerful but are often inapplicable: many deployments expose only predictions (API models, proprietary systems, legacy components), not parameters or intermediate features. Conversely, purely black-box fusion operates only on expert outputs and can struggle under distribution shift: when experts are confidently wrong or mutually ambiguous, no function of their outputs alone can recover the true label. To clarify when each family helps, we identify three idealized data regions: (i) expert-covered—some expert already performs well; (ii) fusion-covered—no single expert suffices, but a function of their outputs does; and (iii) residual—output-only fusion is information-insufficient and one must leverage the raw input.

To address these challenges, we propose **HyperSyn**, an instance-wise synthesis framework that converts expert outputs into the parameters of a target network applied to the input. A hypernetwork consumes the concatenated expert predictions and emits per-example parameters for an encoder; a shared predictor reads its latent representation. In parallel, an expert-outputs-only encoder processes expert outputs to provide a complementary view and a stable anchor. Training balances safety and ambition via two objectives: (i) a hinge oracle-regret term that softly enforces average oracle dominance (defer when experts are right), and (ii) a reliability-weighted distillation that aligns input-aware and outputs-only latents when fusion is reliable, but relaxes alignment when it is

not—enabling the model to go beyond the experts. HyperSyn thus (a) respects the black-box constraint (requires only outputs), (b) strictly contains output-space fusion by injecting input-dependent computation, and (c) provides a loss to trade off safety vs ambition. The main contributions of this paper include:

- We formalize black-box expert fusion with an oracle-dominance objective and introduce a three-region taxonomy—expert-covered, fusion-covered, residual—that explains when selection, output-space fusion, and input-aware synthesis succeed or fail.

- We introduce **HyperSyn**, a hypernetwork-based, instance-wise synthesis method that maps expert outputs to per-example parameters, combining the practicality of black-box aggregation with the expressiveness of parameter-space adaptation.

- We demonstrate strong empirical performance on both synthetic and real-world datasets, showing that HyperSyn achieves state-of-the-art results among black-box expert fusion methods when data exhibit complex and unknown domain structure.

## 2 RELATED WORK

**Ensembling and expert fusion.** Classical ensemble methods—bagging, boosting, and stacking—combine multiple predictors to improve accuracy and robustness (Breiman, 1996; Freund & Schapire, 1996; Friedman, 2001; Ganaie et al., 2022). These approaches typically assume that base learners are trained on similar distributions so that averaging or reweighting their predictions is beneficial. In settings where experts possess complementary expertise (e.g., each excels in a different domain), naive aggregation can be suboptimal or even harmful. A line of recent work, therefore, explores model composition with stronger assumptions on access to experts. In the white-box regime, methods can read or modify internal parameters or representations: HyperMoE introduces hypernetworks that generate "HyperExperts" on the fly using embeddings from unselected experts (Hao Zhao, 2024); WEMoE performs weight-level merging via a mixture-of-experts module to integrate fine-tuned models (Tang et al., 2024). These parameter-space approaches can be powerful but require internals that are often unavailable in practice.

**Black-box expert aggregation.** When only expert outputs are accessible, one must operate in the black-box regime. Simple selectors pick the most confident expert (largest predicted probability) (Pearce et al., 2021; Chen et al., 2023). Mixture-of-Experts (MoE) in this regime learns input-conditioned weights to form a convex combination of expert predictions. Beyond convex mixing, Fusion of Experts (FoE) learns a nonlinear map that fuses expert outputs, enabling gains when information is complementary across experts(Wang et al., 2024). However, purely output-space fusion is information-limited: when expert outputs are ambiguous or uniformly wrong on out-of-domain inputs, no function of the outputs alone can recover the labels.

**Hypernetworks.** Hypernetworks generate the parameters of a target network conditioned on auxiliary inputs, providing a flexible mechanism for parameter sharing and rapid adaptation (Ha et al., 2017). They have been applied to meta-learning (Bertinetto et al., 2016), transfer and modular learning (Ustun et al., 2022; Hao Zhao, 2024), continual learning (von Oswald et al., 2020), and personalization (Ruiz et al., 2024). Our work adapts hypernetworks to the black-box expert setting: HyperSyn conditions the hypernetwork on expert outputs to synthesize an instance-specific predictor that then operates on the raw input. This differentiates HyperSyn from parameter-merging methods (e.g., HyperMoE, WEMoE), which require white-box access, and from output-only fusion (e.g., FoE), which cannot escape output-space bottlenecks on hard out-of-domain inputs.

## 3 PROBLEM FORMULATION

**Overview** We consider the practical scenario where a system must compose a pool of frozen, black–box experts at deployment time. There are three actors: (i) a set of immutable experts $\mathcal{E}$ defined as $m(.)$ that can be queried only for their outputs on an input $x$; (ii) a small labeled reference dataset $\mathcal{D}_{\text{ref}}$ available to learn a *meta-policy*; and (iii) an unknown target distribution $\mathcal{D}_{\text{tgt}}$ on which we ultimately care about risk. Given an input $x$, we can form the expert outputs

$$U(x) \;=\; \big(m_1(x), \ldots, m_K(x)\big) \in \mathbb{R}^{d_u},$$

where $d_u = K \cdot C$ in a $C$-class problem (concatenated class probabilities or logits). The meta-policy $\Pi$ is learned only from $\mathcal{D}_{\text{ref}}$ (without modifying experts) and, at test time, maps $(x, U(x))$ to an instance-conditional action $a_\Pi(x)$ drawn from a chosen decision family $\mathcal{A}$ (e.g., selection, mixture/fusion, or instance-wise synthesis). The induced predictor is

$$M_\Pi(x) \;=\; a_\Pi(x)(x),$$

which may defer to an expert, fuse expert outputs, or apply a synthesized small model to $x$.

**Setup.** Let $\mathcal{X}$ denote the input space and $\mathcal{Y}$ the label space. Given a frozen pool of $K$ experts

$$\mathcal{E} = \{m_i\}_{i=1}^K, \qquad m_i : \mathcal{X} \to \mathcal{P}(\mathcal{Y}),$$

where $\mathcal{P}(\mathcal{Y})$ denotes the prediction space (e.g., logits/probabilities for classification). Each expert exposes only its outputs $m_i(x)$; internal parameters, training data, and intermediate representations are inaccessible. Expert $m_i$ was trained elsewhere on an unknown dataset $\widetilde{\mathcal{D}}_i$, and no further fine-tuning is possible. We are given a labeled reference dataset:

$$\mathcal{D}_{\text{ref}} = \{(x_j, y_j)\}_{j=1}^n \subset \mathcal{X} \times \mathcal{Y},$$

for training a meta-policy, and evaluate on a target distribution $\mathcal{D}_{\text{tgt}}$ over $\mathcal{X} \times \mathcal{Y}$. Fix a per-example loss $\ell : \mathcal{P}(\mathcal{Y}) \times \mathcal{Y} \to \mathbb{R}_{\geq 0}$, and define

$$R(M) \;=\; \mathbb{E}_{(x,y)\sim\mathcal{D}_{\text{tgt}}}\big[\ell(M(x), y)\big], \qquad R_i \;=\; \mathbb{E}_{(x,y)\sim\mathcal{D}_{\text{tgt}}}\big[\ell(m_i(x), y)\big], \qquad R_{\min} \;=\; \min_{i\in[K]} R_i.$$

**Goal: risk minimization with oracle dominance.** Our objective is to construct $M : \mathcal{X} \to \mathcal{P}(\mathcal{Y})$ that achieves low prediction loss while being at least as good as the best frozen expert:

$$\min_M \; R(M) \quad \text{s.t.} \quad R(M) \;\leq\; R_{\min} + \varepsilon, \tag{1}$$

for a small $\varepsilon \geq 0$ (ideally $\varepsilon = 0$). Feasibility is guaranteed by choosing $M = m_{i^\star}$ for some $i^\star \in \arg\min_i R_i$; our aim is to *match or surpass* $R_{\min}$. Because per-example dominance requires ground-truth $y$ at test time, we enforce the constraint in expectation.

**Decision families.** Let $U(x) = (m_1(x), \ldots, m_K(x))$ denote the concatenated outputs of all experts on input $x$, with dimension $d_u = K \cdot C$ in the classification setting ($C = |\mathcal{Y}|$). Let $\mathcal{A}$ be the family of instance-conditional decision rules mapping $x$ to a predictor $a(x) : \mathcal{X} \to \mathcal{P}(\mathcal{Y})$. We highlight four subclasses:

$$\mathcal{A}_{\text{sel}} = \{\, a : \; a(x) = m_{s(x)} \text{ for some } s : \mathcal{X} \to [K] \,\} \qquad \text{(Dynamic Selection)},$$

$$\mathcal{A}_{\text{moe}} = \Big\{\, a : \; a(x) = \sum_{i=1}^K \alpha_i(x)\, m_i(x), \; \alpha(x) \in \Delta^{K-1} \,\Big\} \qquad \text{(Mixture of Experts)},$$

$$\mathcal{A}_{\text{fus}} = \{\, a : \; a(x) = \Phi(U(x)), \; \Phi : \mathbb{R}^{d_u} \to \mathcal{P}(\mathcal{Y}) \,\} \qquad \text{(Fusion)},$$

$$\mathcal{A}_{\text{syn}} = \{\, a : \; a(x) = f_{\theta_x}(x), \; \theta_x = H(U(x)) \,\} \qquad \text{(Instance-wise Synthesis)}.$$

Here $\Delta^{K-1}$ is the probability simplex, $\Phi$ is a meta-predictor acting on expert outputs, and $H : \mathbb{R}^{d_u} \to \mathbb{R}^{d_\theta}$ is a hypernetwork that maps expert outputs $U(x)$ to parameters $\theta_x$ of a synthesized target model $f_{\theta_x}$. Dynamic selection and MoE are special cases of fusion: $\mathcal{A}_{\text{sel}} \subseteq \mathcal{A}_{\text{fus}}$ (take $\Phi$ to route one expert), and $\mathcal{A}_{\text{moe}} \subseteq \mathcal{A}_{\text{fus}}$ (take $\Phi$ to form convex combinations in output space). Fusion $\mathcal{A}_{\text{fus}}$ permits nonlinear combinations of expert outputs, remaining strictly output-space. Instance-wise synthesis $\mathcal{A}_{\text{syn}}$ operates in parameter space (via $\theta_x$), generally offering a different and often richer function class than output-space fusion as shown in appendix A.3.

A meta-policy $\Pi$ (learned from $\mathcal{D}_{\text{ref}}$) selects, for each $x$, an action $a_\Pi(x) \in \mathcal{A}$, yielding the predictor $M_\Pi(x) = a_\Pi(x)(x)$. The problem is therefore the constrained risk minimization in equation 1 over a chosen hypothesis class of meta-policies (e.g., gating networks, stackers, hypernetworks, or hybrids). Different choices of $\mathcal{A}$ and recover selection, MoE, and instance-wise synthesis; all are trained on $\mathcal{D}_{\text{ref}}$ and evaluated on $\mathcal{D}_{\text{tgt}}$.

The problem encountered three challenges: (i) Black-box experts-Due to many real-world situations, we only observe expert outputs; internal weights and parameters are inaccessible. Methods

must leverage signals (probabilities, confidences, disagreements) rather than parameters. (ii) Out-of-domain inability.-Experts may fail on target domains. Simple fusion often inherits these failures; methods must extrapolate beyond expert support.(iii) Safety vs ambition-On the one hand, we need oracle-dominance to protect against degradations relative to the expert pool; on the other hand, we need the model to improve beyond the experts when the reference labels reveal improvements.

### 3.1 A Three–Way Taxonomy of Data Regions

To illustrate the challenge of out-of-domain inability, we could partition the data space into three regions. define the oracle per-example loss

$$\ell^\star(x,y) = \min_{i\in[K]} \ell(m_i(x), y).$$

We define regions relative to a family $\mathcal{F}$ of fusion functions on expert outputs (e.g. linear stacking, shallow MLPs) $g : \mathcal{P}(\mathcal{Y})^K \to \mathcal{P}(\mathcal{Y})$:

**(1) Expert–covered region $\Omega_{\text{exp}}$.** There exists an expert that is already (nearly) optimal on $(x,y)$. For a tolerance $\tau > 0$, define

$$\Omega_{\text{exp}} = \big\{(x,y) \in \mathcal{X} \times \mathcal{Y} : \ell^\star(x,y) \leq \tau\big\}, \qquad \ell^\star(x,y) = \min_{i\in[K]} \ell\big(m_i(x), y\big).$$

Thus, at $(x,y)$ at least one expert attains loss at most $\tau$; exact selection achieves zero per-example regret on this set.

**(2) Fusion–covered region $\Omega_{\text{fus}}(\mathcal{F})$.** No single expert suffices, yet a *function of their outputs* can predict well. Given a function class $\mathcal{F}$ acting on expert outputs $U(x)$, define

$$\Omega_{\text{fus}}(\mathcal{F}) = \big\{(x,y) : \ell^\star(x,y) > \tau \text{ and } \exists g \in \mathcal{F} \text{ s.t. } \ell\big(g(U(x)), y\big) \leq \tau\big\}.$$

Here the experts carry complementary information; a learned non-linear fusion of their outputs can recover the label even though each expert (and sometimes any convex mixture of their outputs) fails.

**(3) Residual region $\Omega_{\text{res}}(\mathcal{F})$.** Neither any single expert nor any function of the outputs suffices:

$$\Omega_{\text{res}}(\mathcal{F}) = \big\{(x,y) : \ell^\star(x,y) > \tau \text{ and } \forall g \in \mathcal{F}, \ \ell\big(g(U(x)), y\big) > \tau\big\}.$$

On this set, the expert outputs are information–insufficient for predicting $y$; one must leverage additional signal from $x$. We could prove the limitations of existing methods using those data regions. Without loss of generality, assuming the task is binary classification, multiclass follows by one-vs-rest arguments. And suppose the loss $\ell$ is cross entropy. And we could define the residual set with margin $A_\gamma^+$ and $A_\gamma^-$ in the appendix A.2.

**Theorem 1.** *Let $\widehat{p}$ be any predictor produced by (i) dynamic selection $x \mapsto m_{s(x)}(x)$ or (ii) a convex MoE $\widehat{p}(x) = \sum_{i=1}^K \alpha_i(x)\, m_i(x)$ with $\alpha(x) \in \Delta^{K-1}$. For any $\gamma \in (0, \frac{1}{2})$,*

$$\mathbb{E}\big[\ell(\widehat{p}(X), Y)\big] \geq \mathbb{P}\big((X,Y) \in A_\gamma\big) \cdot \big(-\log(\tfrac{1}{2} - \gamma)\big).$$

On the residual set $A_\gamma$ where all experts are confidently wrong, every selection/MoE predictor must misclassify and incur a loss at least $-\log(\frac{1}{2} - \gamma)$ per point.

**Theorem 2.** *Suppose we have two hypothesis classes for $U(x)$ and $x$,*

$$\mathcal{H}_{\text{out}} := \big\{ x \mapsto g_u(U(x)) : g_u : \text{Range}(U) \to [0,1] \big\}, \qquad \mathcal{H}_{\text{in}} := \big\{ x \mapsto g_x(x) : g_x : \mathcal{X} \to [0,1] \big\}.$$

*(i) The function from $\mathcal{H}_{\text{in}}$ dominates $\mathcal{H}_{\text{out}}$*

$$\inf_{f\in\mathcal{H}_{\text{in}}} \mathbb{E}\big[\ell(f(X), Y)\big] \leq \inf_{f\in\mathcal{H}_{\text{out}}} \mathbb{E}\big[\ell(f(X), Y)\big].$$

*(ii) Strict inequality under $U$–collisions.* *Suppose there exist disjoint sets $A, B \subseteq \mathcal{X}$ with $\alpha := \Pr(X \in A) > 0$ and $\beta := \Pr(X \in B) > 0$ such that*

$$U(x) = u_0 \quad \text{for all } x \in A \cup B, \qquad \pi_A := \Pr(Y{=}1 \mid X \in A) \neq \pi_B := \Pr(Y{=}1 \mid X \in B).$$

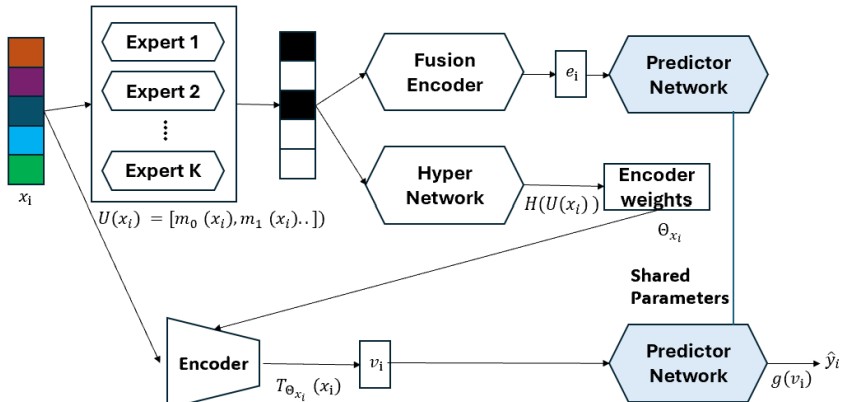

Figure 1: Overview of HyperSyn.

*Let $h(t) = -t \log t - (1-t) \log(1-t)$ be the entropy and set $\bar{\pi} := \frac{\alpha \pi_A + \beta \pi_B}{\alpha + \beta}$. Then the optimal risks over the two classes satisfy*

$$\inf_{f \in \mathsf{H}_{\text{out}}} \mathbb{E}\big[\ell(f(X), Y)\big] - \inf_{f \in \mathsf{H}_{\text{in}}} \mathbb{E}\big[\ell(f(X), Y)\big] \geq \alpha \, \text{KL}(\pi_A \| \bar{\pi}) + \beta \, \text{KL}(\pi_B \| \bar{\pi}) > 0 \,.$$

It shows the inability of FoE model in the residual region. The proofs are described in appendix A.5.

## 4 PROPOSED METHOD

We proposed HyperSyn, a deep learning framework that is able to address the challenges described above. HyperSyn follows the instance-wise synthesis paradigm $a(x) = f_{\theta_x}(x)$ with per-example parameters $\theta_x$ produced by a hypernetwork conditioned only on $U(x)$, therefore, it is applicable for a blackbox expert setting. And it is more expressive for attacking the out of domain issues.

### 4.1 OVERVIEW OF HYPERSYN

As illustrated in figure 1, given the $x_i \in \mathbb{R}^d$ which is the $i$-th data point and we have the concatenation of expert outputs $U(x_i) = (m_1(x_i), \ldots, m_K(x_i))$ , we feed them to HyperSyn which contains Hypernetwork, Target Encoder Network, Fusion Encoder and Shared predictor.

### 4.2 HYPERNETWORK FOR SYNTHESIZING INPUT ENCODERS

We design the *hypernetwork* and the *Target Encoder Network* as follows:

- **Hypernetwork** $H : \mathbb{R}^{d_u} \to \mathbb{R}^{d_\theta}$ is an MLP that takes flattened expert outputs $U(x_i) \in \mathbb{R}^{d_u}$ as the input and outputs a parameter vector $\theta_{x_i} = H(U(x_i)) \in \mathbb{R}^{d_\theta}$ for a target network.
- **Target Encoder Network** $T_{\theta_{x_i}}(x_i) : \mathbb{R}^d \to \mathbb{R}^h$ which is another MLP formed by *reshaping* the parameter vector $\theta_{x_i}$ into its weight matrices and the bias vectors. The network transforms the raw input to a latent representation $v(x) = T_{\theta_x}(x) \in \mathbb{R}^h$,

### 4.3 FUSION ENCODER

It is an MLP $e(x_i) = E(U(x_i)) \in \mathbb{R}^h$ that map the flattened expert outputs to latent representation. Passing $e(x)$ through the shared predictor $g$ yields an auxiliary prediction $\hat{y}^u(x_i) = g(e(x_i))$. This branch never sees $x$; it is a function of $U(x_i)$ and acts as a learned surrogate of the expert output.

### 4.4 SHARED PREDICTOR

A shared predictor $g : \mathbb{R}^h \to \mathbb{R}^C$ (or $\mathbb{R}$ in binary) maps either $v(x)$ *or* embedding $e(x)$ to prediction logits. Sharing the predictor has the advantages, including:(i) **Reduce Parameters Generated by**

**Hypernetwork**- Avoid generating whole experts by hypernetwork for different data points by focusing on partial parameters so that the learning becomes easier. (ii) **Unified latent space for two view points**- For two view points - the expert output concatenation $U$ and input $x$, we have different representation encoder. The shared predictor forces them to be in the same space so that the model can benefit from complementary information about them and help the distillation of knowledge.

### 4.5 POTENTIAL EXTENSION TO COMPLEX ARCHITECTURE

Conceptually, HyperSyn is orthogonal to the base architecture: the hypernetwork only needs to generate some encoder parameters conditioned on U(x), such as ViT-based image experts, small LLM experts for text, and the shared predictor can be any differentiable head.

### 4.6 TRAINING OBJECTIVES

Let $\hat{y}(x) = g\big(T_{\theta_x}(x)\big)$ denote the input based prediction (logits), and let $\hat{y}^{\mathrm{u}}(x) = g\big(E(U(x))\big)$ denote the expert output based prediction. Write $\ell(\cdot, y)$ for per-example supervised loss. Define the per–example oracle loss $\ell^\star(x, y) = \min_{i \in [K]} \ell(m_i(x), y)$ computed directly from black–box outputs. At test time, the final prediction is always $\hat{y}(x)$ from the input-aware branch; the outputs-only branch is auxiliary and is used only during training. We have the following loss:

#### 4.6.1 SUPERVISED LOSS

We have two supervised losses: $\ell(\hat{y}, y)$ which is the main prediction that take $x$ as input to fit labels and $\ell(\hat{y}^{\mathrm{u}}, y)$ which compute prediction loss based on $U(x)$ using cross entropy.

#### 4.6.2 ORACLE REGRET LOSS

It encourages oracle dominance to balance the Safety vs. ambition challenge.

$$\mathcal{L}_{\mathrm{regret}} = \max\big\{\,\ell(\hat{y}(x), y) - \ell^\star(x, y),\, 0\,\big\} \tag{2}$$

It is a hinge loss that penalized only when $\ell(\hat{y}, y)$ exceeds $\ell^\star(x, y)$; this softly enforces $R(M) \leq R_{\min}$ on average and protects against degradation relative to the pool. Therefore, it helps the HyperSyn to follow the experts when they are correct to capture the Expert-covered region $\Omega_{\mathrm{exp}}$, and also allow it go beyond the experts when they fail.

#### 4.6.3 FUSION DISTILLATION LOSS

To encourage the model to capture the Fusion-covered region $\Omega_{\mathrm{fus}}(\mathcal{F})$, we align the input aware latent $v(x_i)$ with the expert outputs aware latent $e(x_i)$:

$$\mathcal{L}_{\mathrm{dist}} = w(x)\,\|v(x) - e(x)\|_2^2, \qquad w(x) \propto \frac{1}{\ell(\hat{y}^{\mathrm{u}}(x), y) + \epsilon} \tag{3}$$

The intuition is that when the fusion encoder is reliable (low loss, probably indicates it is in Fusion-covered region), the model trusts $E(U(x))$ and pulls $v(x)$ closer; when it is unreliable(eg, in Residual region $\Omega_{\mathrm{res}}(\mathcal{F})$), the weight decays and the model leans on $x \mapsto T_{\theta_x}(x)$.

#### 4.6.4 TOTAL LOSS

The total loss is computed as the sum of above loss with parameter regularization:

$$\mathcal{L} = \ell(\hat{y}, y) + \ell(\hat{y}^{\mathrm{u}}, y) + \lambda_{\mathrm{reg}}\mathcal{L}_{\mathrm{regret}} + \lambda_{\mathrm{dist}}\mathcal{L}_{\mathrm{dist}} + \rho\,\|\theta_x\|_2^2 \tag{4}$$

where $\lambda_{\mathrm{reg}}, \lambda_{\mathrm{dist}}, \rho$ are the loss weights.

## 5 EXPERIMENT SETUP

### 5.1 SYNTHETIC PIECEWISE XOR (PXOR) DATASET

We focus on tabular, multi-expert settings with controllable distribution shifts. So we construct *Piecewise XOR (PXOR)* dataset. Each example is drawn from $[-1, 1]^3$, with the third dimension $x_3$ acting as a switch: when $x_3 > 0$ the label follows an OR rule using $(x_1 > 0) \lor (x_2 > 0)$, and when $x_3 \leq 0$ it follows the standard XOR rule $(x_1 \cdot x_2 < 0)$ instead. It can recover to $XOR$ by just setting $x_3 \leq 0$. We also set two rule-based experts, one producing high probabilities when $x_1 > 0$ and the other $x_2 > 0$. Simple ensembling that individually attends to $x_1$ or $x_2$ performs well in OR regions but fails in XOR regions, and the fusion of two experts can tackle XOR. The details of the data are described in appendix A.8

### 5.2 SYNTHETIC MULTI–DOMAIN MULTI–FUNCTION (MDMF) DATASET

To evaluate the ability of HyperSyn under controlled yet challenging conditions, we construct a synthetic dataset called *Multi–Domain Multi–Function (MDMF)*. In MDMF, each of the $K$ domains is defined by its own Gaussian center in input space and a distinct linear function that labels points according to $y = \mathbb{K}\{w_k^\top x > 0\}$. Training, validation, and test splits are balanced across domains, yielding a binary classification problem where the global decision boundary is a patchwork of domain-specific linear separators. As a result, the dataset exhibits both covariate shift (different input clusters) and functional shift (different label rules), which cannot be captured by a single linear model. The rationale is to emulate realistic settings where frozen experts are trained on different domains and thus expose heterogeneous decision rules. The details are described in appendix A.9.

### 5.3 REAL WORLD DATASETS

We also conduct experiments on 9 real word data sets that cover a variety of domains. Specifically, the data sets include adult(Kohavi, 1996), mushroom(Lincoff, 1981), gesture(Madeo & Peres, 2013), mice_protein(Higuera et al., 2015), har(Anguita et al., 2013), isolet(Cole & Fanty, 1991), eucalyptus(Bulluch, 1992), vowel(Deterding & Robinson, 1988) and gas_drift(Vergara, 2012). The details are described in Appendix A.11.

### 5.4 BASELINES

We compared HyperSyn against commonly used and recent blackbox methods that exploit only the outputs of experts. The comparison includes Oracle Expert, where the expert with the best accuracy for the data points is selected as output. It is treated as the ideal but not realistic baseline, as we assume the data labels are available in the test data to select the oracle. The other baselines includes Best Expert, Most Confidence((Pearce et al., 2021; Chen et al., 2023), average of experts(MoE-AVG), gated mixture of expert(MoE-Gated)(Jacobs et al., 1991; Jordan & Jacobs, 1994; Masoudnia & Ebrahimpour, 2014)and FoE(Wang et al., 2024) which which consume U(x) as input. And we also add input-augmented baseline FoE+X which consume both x and U(x) same as HyperSyn, the details of baselines is listed in appendix A.10. The parameter setting of HyperSyn is described in A.4.

## 6 RESULTS

### 6.1 SYNTHETIC DATA ANALYSIS

Table 1 shows the results of different methods on the synthetic data set. For XOR dataset, HyperSyn obtains better accuracy than the others while Fusion of expert(FoE) achieves the second best, which is expected as the data set is a kind of fusion-covered data region where the fusion of experts could predict the label well. When come to the piecewise XOR dataset, HyperSyn is only worse than the Oracle Expert, but FoE achieves the worst performance as the additional feature changes the fusion behaviour of XOR, which implies the residual region where the FoE model fails. For MDMF dataset. HyperSyn outperforms the baselines. And HyperSyn is marginally better than FoE+X(p-value=0.1804) and also higher than other methods that rely only on expert outputs

Table 1: Comparison of methods on three synthetic datasets: XOR, PXOR, and MDMF.

| Method | XOR | PXOR | MDMF |
|---|---|---|---|
| Oracle Expert | 0.7462±0.0117 | 0.8744±0.0100 | 0.7357±0.2365 |
| Best Expert | 0.5032±0.0139 | 0.6314±0.0031 | 0.5356±0.1019 |
| Most Confident | 0.7462±0.0117 | 0.8744±0.0100 | 0.4996±0.1031 |
| MoE-Avg | 0.5500±0.0125 | 0.6782±0.0096 | 0.4809±0.1288 |
| MoE-Gated | 0.5092±0.0131 | 0.6370±0.0088 | 0.4951±0.1242 |
| FoE | 0.7528±0.0059 | 0.6208±0.0113 | 0.5882±0.0935 |
| FoE+X | 0.7120±0.0245 | 0.6638±0.0233 | 0.7768±0.1384 |
| wo Regret | 0.9258±0.0367 | 0.9434±0.0135 | 0.7966±0.1245 |
| wo Distillation | 0.9342±0.0230 | 0.9102±0.0124 | 0.7951±0.1291 |
| HyperSyn | **0.9362±0.0758** | **0.9456±0.0511** | **0.7985±0.1274** |

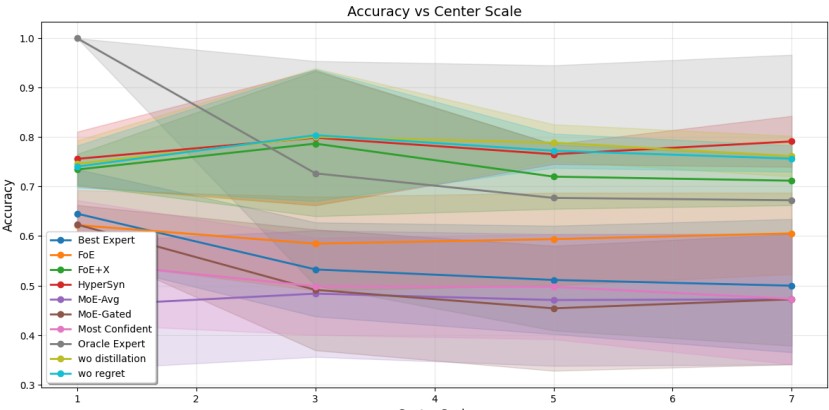

Figure 2: Center scale versus accuracy for MDMF data

by a large margin. Notably, it is about 6% improvement, which is significally higher than Oracle expert(p-value=0.0142<0.05), showing that instance-wise synthesis can go beyond the pool when expert coverage is incomplete.

Figure 2 plots the different center scale(the variance of the center of domains, or the domain separation) versus accuracy. We observe that the HyperSyn win the other methods more obviouly when the center scale is high. As the domain becomes more separate, the more spread of different center of domains and using experts to predict their unknwon domains become more difficult, , matching the intuition that larger covariate shift increases the prevalence of residual regions where output-only fusion is information-limited and input-aware synthesis is needed.

Table 2: Comparison of methods across the first five datasets. Reported are mean accuracy ± standard deviation (5 runs). Best non-Oracle method per dataset is boldfaced.

| Method | Adult | Mushroom | Gesture | Mice protein | HAR |
|---|---|---|---|---|---|
| Oracle Expert | 0.9020±0.0055 | 0.9994±0.0007 | 0.7572±0.0142 | 0.9741±0.0320 | 0.9641±0.0716 |
| Best Expert | 0.8503±0.0019 | 0.8167±0.0052 | 0.2868±0.0148 | 0.2407±0.0285 | 0.2362±0.0481 |
| Most Confident | 0.8184±0.0066 | 0.5081±0.0043 | 0.4373±0.0151 | 0.1380±0.0446 | 0.1331±0.0244 |
| MoE-Avg | 0.8224±0.0023 | 0.8281±0.0123 | 0.4629±0.0128 | 0.2843±0.0753 | 0.4403±0.0772 |
| MoE-Gated | 0.8495±0.0044 | 0.9334±0.0125 | 0.4049±0.0530 | 0.3037±0.0944 | 0.5518±0.1311 |
| FoE | 0.8487±0.0031 | 0.9369±0.0090 | 0.4792±0.0088 | 0.5028±0.0569 | 0.9250±0.0081 |
| FoE+X | 0.7616±0.0018 | 0.9755±0.0063 | 0.4822±0.0080 | 0.5444±0.0677 | **0.9435±0.0101** |
| wo Regret | 0.8487±0.0024 | **0.9909±0.0099** | **0.4890±0.0047** | 0.5556±0.0539 | 0.9384±0.0105 |
| wo Distillation | 0.8486±0.0031 | 0.9867±0.0097 | 0.4753±0.0073 | 0.5037±0.0746 | 0.9247±0.0180 |
| HyperSyn | **0.8583±0.0028** | 0.9906±0.0085 | 0.4835±0.0078 | **0.5639±0.0503** | 0.9417±0.0125 |

Table 3: Comparison of methods across the remaining four datasets.

| Method | Eucalyptus | Gas Drift | Vowel | Isolet |
|---|---|---|---|---|
| Oracle Expert | 0.8054±0.0361 | 0.9353±0.0050 | 0.7919±0.0252 | 0.9349±0.0169 |
| Best Expert | 0.3703±0.0324 | 0.7042±0.0084 | 0.2051±0.0191 | 0.1031±0.0156 |
| Most Confident | 0.3581±0.0410 | 0.5898±0.0278 | 0.0990±0.0444 | 0.0254±0.0193 |
| MoE-Avg | 0.4284±0.0388 | 0.6423±0.0126 | 0.2495±0.0442 | 0.1147±0.0226 |
| MoE-Gated | 0.3486±0.0416 | 0.7070±0.0058 | 0.2394±0.1187 | 0.2233±0.0637 |
| FoE | 0.4635±0.0357 | 0.8453±0.0119 | 0.3121±0.0206 | 0.6451±0.0231 |
| FoE+X | 0.2811±0.0189 | 0.2087±0.0125 | 0.3778±0.0453 | 0.7700±0.0117 |
| wo Regret | 0.5189±0.0411 | 0.8902±0.0131 | 0.3949±0.0716 | **0.8053±0.0156** |
| wo Distillation | 0.4973±0.0393 | 0.8826±0.0205 | 0.4071±0.0297 | 0.7890±0.0106 |
| HyperSyn | **0.5257±0.0499** | **0.8954±0.0123** | **0.4111±0.0354** | 0.8012±0.0260 |

## 6.2 REAL WORLD DATA ANALYSIS

Table 2 and 3 show the comparison of different methods in the 9 real world datasets. HyperSyn achieves the best accuracy(excluding the Oracle expert and variants) among 8 datasets. For the remaining HAR dataset, the accuracy of HyperSyn and the best methods are not statistically different. HyperSyn is also consistently better than the baselines in 4 datasets, for example, it achieves 0.8012, which is statistically better than the second best FoE+X 0.77(p-value=$0.0301 < 0.05$) in isolet data. And HyperSyn is also marginally better than the second best in vowel dataset(p-value=0.1398). The results show that HyperSyn is generally better than existing blackbox models due to its instance-wise synthesis nature.

## 6.3 ABLATION STUDY

We conduct the ablation study to evaluate the effectiveness of different components of HyperSyn. In the experiments, we introduce the following variants: (i)**wo Regret**-where the oracle regret in the loss 2 is removed, so the model depends less on the experts. (ii) **wo Distillation**-where the distillation loss 3 is removed, so the model learn less on the fusion view of expert outputs. As shown in table 1, for XOR, PXOR and MDMF dataset, HyperSyn, wo Regret, and wo Distillation do not show a significant difference, which is expected as the data does not contain a fusion-covered region, which hinders the use of distillation. And the experts are weak on predicting out of domains, which makes the oracle regret less useful. In Table 2 and 3, after removing regret loss, 5 datasets show decreased performance, and for the remaining 4 datasets show an insignificant difference. And after removing the Distillation loss, all data sets have decreased accuracy.

## 7 CONCLUSION

In this paper, we introduced *HyperSyn*, a framework that leverages pretrained expert models by synthesizing instance-specific predictors through a hypernetwork conditioned on expert outputs. Unlike prior methods that require white-box access, HyperSyn operates naturally in the black-box setting where only the outputs of experts are available. To characterize the limitations of existing approaches, we identified three types of data regions and showed that HyperSyn is capable of addressing all of them. We further proposed an oracle-regret loss to balance reliance on existing experts with the ability to go beyond them when encountering new domains. Extensive experiments on both synthetic and real-world datasets demonstrate that HyperSyn consistently outperforms existing black-box expert fusion methods, particularly when data exhibit diverse domain and function regions. For future work, extending HyperSyn to generative tasks (e.g., text or image synthesis) would broaden its applicability. Another promising direction is exploring the "frugal" regime, where the goal is to maintain high performance while relying on a smaller number of experts, thereby improving efficiency and scalability.

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

# A APPENDIX

## A.1 USAGE OF LARGE LANGUAGE MODEL

We use LLM for polishing the writing, including checking grammar.

## A.2 DEFINITION OF RESIDUAL SET

**Definition A.1** (Residual set with margin). *Fix $\gamma \in (0, \frac{1}{2})$. Define the event*

$$A_\gamma^+ := \big\{(x,y) : y=1, \ \max_{i\in[K]} m_i(x) \leq \tfrac{1}{2} - \gamma\big\}, \quad A_\gamma^- := \big\{(x,y) : y=0, \ \min_{i\in[K]} m_i(x) \geq \tfrac{1}{2} + \gamma\big\},$$

*and $A_\gamma := A_\gamma^+ \cup A_\gamma^-$. On $A_\gamma$ every expert is confidently wrong by a margin $\gamma$.*

## A.3 HYPERSYN CONTAINS OUTPUT-ONLY FUSION

Recall that we consider a set of black-box experts $\{m_k\}_{k=1}^K$ and denote by $U(x) \in \mathbb{R}^{d_U}$ the concatenated expert outputs (logits or probabilities) on input $x$. We define two hypothesis classes:

$$\mathcal{A}_{\text{fus}} := \Big\{ f : x \mapsto F\big(U(x)\big) \mid F \in \mathcal{F}\Big\},$$

$$\mathcal{A}_{\text{syn}} := \Big\{ f : x \mapsto g\big(T_{\theta_x}(x)\big) \mid \theta_x = H\big(U(x)\big), \ H \in \mathcal{H}, \ T \in \mathcal{T}, \ g \in \mathcal{G}\Big\},$$

where $\mathcal{F}$ is a class of fusion networks acting only on $U(x)$, $\mathcal{H}$ is a hypernetwork class mapping $U(x)$ to encoder parameters $\theta_x$, $\mathcal{T}$ is a class of encoders $T_{\theta_x}$ acting on the raw input $x$, and $\mathcal{G}$ is a class of predictors acting on latent representations.

Intuitively, $\mathcal{A}_{\text{fus}}$ consists of all predictors that operate only on expert outputs, whereas $\mathcal{A}_{\text{syn}}$ consists of predictors that can revisit $x$ in an input-dependent way, conditioned on $U(x)$ through the hypernetwork.

**Containment $\mathcal{A}_{\text{fus}} \subseteq \mathcal{A}_{\text{syn}}$.** We show that any output-only fusion model can be simulated by a suitable choice of HyperSyn components. For clarity, we consider the common case where the encoder $T_{\theta_x}$ is implemented as an MLP, but the argument only requires that $T_{\theta_x}$ can ignore its input $x$ and use its parameters $\theta_x$ to produce a latent vector.

**Lemma 1** (Fusion is a special case of synthesis). *Assume that the composition $g \circ T_\theta$ is a universal approximator for functions of the encoder parameters $\theta$, in the sense that for any continuous $F : \mathbb{R}^{d_U} \to \mathcal{Y}$ and any $\varepsilon > 0$ there exist $H \in \mathcal{H}, T \in \mathcal{T}, g \in \mathcal{G}$ such that*

$$\sup_{u \in \mathbb{R}^{d_U}} \big\| g\big(T_{H(u)}(x_{\text{ref}})\big) - F(u)\big\| \leq \varepsilon$$

*for some fixed reference input $x_{\text{ref}}$ (independent of $u$). Then for any $f \in \mathcal{A}_{\text{fus}}$ and any $\varepsilon > 0$ there exists $\tilde{f} \in \mathcal{A}_{\text{syn}}$ such that $\sup_x \|f(x) - \tilde{f}(x)\| \leq \varepsilon$.*

*Proof.* Take any fusion model $f \in \mathcal{A}_{\text{fus}}$. By definition, there exists $F \in \mathcal{F}$ such that

$$f(x) = F\big(U(x)\big) \quad \text{for all } x.$$

We construct a HyperSyn model that reproduces this mapping.

Consider an encoder $T_\theta(x)$ with the following property: it can be parameterized such that its dependence on $x$ is removed. For instance, in an MLP encoder we can set all weights on the input $x$ to zero and use only bias terms (or a constant input vector) to produce a latent representation. Formally, fix a reference input $x_{\text{ref}}$ and choose $T_\theta$ so that

$$v(x) = T_\theta(x) = T_\theta(x_{\text{ref}}) \quad \text{for all } x.$$

Thus $v(x)$ depends only on $\theta$, not on the actual $x$.

Next, let the hypernetwork $H$ map the fusion input $u = U(x)$ to parameters $\theta_x = H(u)$. Since $g \circ T_\theta$ is a universal approximator in $\theta$, there exist $H, T, g$ such that

$$g\big(T_{H(u)}(x_{\text{ref}})\big) \approx F(u) \quad \text{for all } u.$$

Defining

$$\tilde{f}(x) := g\big(T_{\theta_x}(x)\big) = g\big(T_{H(U(x))}(x_{\text{ref}})\big),$$

we obtain

$$\tilde{f}(x) \approx F\big(U(x)\big) = f(x) \quad \text{for all } x.$$

Intuitively, we have made the encoder ignore the raw input $x$ and use its hypernetwork-generated parameters to encode only the expert outputs $U(x)$; the predictor $g$ then plays the role of the original fusion network $F$. Given sufficient capacity, the approximation can be made arbitrarily close, yielding the claimed inclusion. $\square$

In other words, HyperSyn can simulate any pure output-only fusion pipeline by choosing the encoder to ignore $x$ and letting the hypernetwork carry all dependence on $U(x)$.

### A.4 PARAMETER SETTING

The latent dimension $h$ of the hypernetwork, target encoder network, and fusion encoder is chosen as $d/4$ with single-layer MLP. We also set $\mathcal{L}_{\text{regret}} = 0.01$, $\mathcal{L}_{\text{dist}} = 0.1$ and $\rho = 0.0001$. We applied ADAM optimizer with the learning rate $1 \times 10^{-3}$ and the batch size of 128. We split the data randomly with 5% as reference data, and the remaining is split into 80% training data for experts and 20% test data. Each expert is a two-layer MLP, and 50% of domains are used for expert training; in other words, no experts can predict well on the remaining 50% of unknown domains. We perform training and testing for five runs in our experiments. All the performance comparison is based on the accuracy of the test data, and the standard deviation among the five runs is also presented. All experiments are conducted on a server with four NVidia Tesla V100-PCIE-32GB GPU, 250 GB memory, and Intel(R) Silver 4114 CPU.

### A.5 PROOFS OF THEOREM

*Theorem 1.* Fix $(x,y) \in A_\gamma^+$. Then $y=1$ and $m_i(x) \leq \frac{1}{2} - \gamma$ for all $i$. For selection, $\widehat{p}(x) = m_{s(x)}(x) \leq \frac{1}{2} - \gamma$, hence $\ell(\widehat{p}(x), 1) = -\log \widehat{p}(x) \geq -\log(\frac{1}{2} - \gamma)$. For a convex MoE $\widehat{p}(x) = \sum_i \alpha_i(x) m_i(x)$ with $\alpha(x) \in \Delta^{K-1}$, convexity gives $\widehat{p}(x) \leq \frac{1}{2} - \gamma$, so the same bounds hold.

Now fix $(x,y) \in A_\gamma^-$. Then $y=0$ and $m_i(x) \geq \frac{1}{2} + \gamma$ for all $i$. For selection, $\widehat{p}(x) \geq \frac{1}{2} + \gamma$, hence $\ell(\widehat{p}(x), 0) = -\log(1 - \widehat{p}(x)) \geq -\log(\frac{1}{2} - \gamma)$. For a convex MoE, $\widehat{p}(x) \geq \frac{1}{2} + \gamma$ by convexity, yielding the same bound. Taking expectations over $(X, Y)$ and noting that outside $A_\gamma$ the losses are nonnegative proves the two claimed inequalities. $\square$

**Lemma 2.** *Let $\alpha, \beta > 0$, $\pi_A, \pi_B \in (0, 1)$, and define the mixture parameter*

$$\bar{\pi} := \frac{\alpha \pi_A + \beta \pi_B}{\alpha + \beta}.$$

*With the entropy $h(t) = -t \log t - (1-t) \log(1-t)$ and the $\mathrm{KL}(\pi \| q) = \pi \log \frac{\pi}{q} + (1-\pi) \log \frac{1-\pi}{1-q}$,*

$$\alpha \mathrm{KL}(\pi_A \| \bar{\pi}) + \beta \mathrm{KL}(\pi_B \| \bar{\pi}) = (\alpha + \beta) h(\bar{\pi}) - \alpha h(\pi_A) - \beta h(\pi_B).$$

*Proof.* Start from the left-hand side and expand each KL term:

$$\alpha \mathrm{KL}(\pi_A \| \bar{\pi}) + \beta \mathrm{KL}(\pi_B \| \bar{\pi}) = \alpha \left[ \pi_A \log \frac{\pi_A}{\bar{\pi}} + (1 - \pi_A) \log \frac{1 - \pi_A}{1 - \bar{\pi}} \right] + \beta \left[ \pi_B \log \frac{\pi_B}{\bar{\pi}} + (1 - \pi_B) \log \frac{1 - \pi_B}{1 - \bar{\pi}} \right]$$

$$= \alpha \Big[ \pi_A \log \pi_A + (1 - \pi_A) \log(1 - \pi_A) \Big] + \beta \Big[ \pi_B \log \pi_B + (1 - \pi_B) \log(1 - \pi_B) \Big]$$

$$- \Big[ (\alpha \pi_A + \beta \pi_B) \log \bar{\pi} + (\alpha(1 - \pi_A) + \beta(1 - \pi_B)) \log(1 - \bar{\pi}) \Big].$$

Recognize the weighted component entropies in the first line:

$$\alpha \big[ \pi_A \log \pi_A + (1 - \pi_A) \log(1 - \pi_A) \big] = -\alpha h(\pi_A), \qquad \beta \big[ \pi_B \log \pi_B + (1 - \pi_B) \log(1 - \pi_B) \big] = -\beta h(\pi_B).$$

For the coefficients of $\log \bar{\pi}$ and $\log(1 - \bar{\pi})$, use the definition of the mixture:

$$\alpha \pi_A + \beta \pi_B \;=\; (\alpha + \beta)\bar{\pi}, \qquad \alpha(1 - \pi_A) + \beta(1 - \pi_B) \;=\; (\alpha + \beta)\,(1 - \bar{\pi}).$$

Hence

$$\alpha \,\mathrm{KL}(\pi_A \| \bar{\pi}) + \beta \,\mathrm{KL}(\pi_B \| \bar{\pi}) = -\alpha h(\pi_A) - \beta h(\pi_B) - \Big[(\alpha + \beta)\bar{\pi}\log\bar{\pi} + (\alpha + \beta)(1 - \bar{\pi})\log(1 - \bar{\pi})\Big]$$
$$= -\alpha h(\pi_A) - \beta h(\pi_B) + (\alpha + \beta)\,h(\bar{\pi}),$$

which is precisely the claimed identity. $\qquad\square$

*Theorem 2. (i) The function from $\mathcal{H}_{\mathrm{in}}$ dominates $\mathcal{H}_{\mathrm{out}}$* By construction, every $f \in \mathsf{H}_{\mathrm{out}}$ is a special case of a map $\mathcal{X} \to [0,1]$, for example, we could construct f(U(X)). Hence $\mathsf{H}_{\mathrm{out}} \subseteq \mathsf{H}_{\mathrm{in}}$, so

$$\inf_{f \in \mathsf{H}_{\mathrm{in}}} \mathbb{E}[\ell(f(X), Y)] \;\leq\; \inf_{f \in \mathsf{H}_{\mathrm{out}}} \mathbb{E}[\ell(f(X), Y)].$$

*(ii) Strict inequality.* Since $U(x)$ is constant on $A \cup B$, every $f \in \mathsf{H}_{\mathrm{out}}$ must predict a *single* probability $p \in [0,1]$ on $A \cup B$. Its contribution to the expected cross entropy on $A \cup B$ therefore equals

$$L_{\mathrm{out}}(p) \;=\; \alpha\, H(\pi_A, p) \;+\; \beta\, H(\pi_B, p), \qquad H(\pi, p) := -\pi \log p - (1 - \pi)\log(1 - p).$$

Using the decomposition $H(\pi, p) = h(\pi) + \mathrm{KL}(\pi \| p)$ and the convexity of $p \mapsto \mathrm{KL}(\pi \| p)$, the unique minimizer is obtained by setting the derivative to zero:

$$L'_{\mathrm{out}}(p) \;=\; -\frac{\alpha \pi_A + \beta \pi_B}{p} + \frac{\alpha(1 - \pi_A) + \beta(1 - \pi_B)}{1 - p} \;=\; 0 \quad \Longrightarrow \quad p^\star = \bar{\pi} := \frac{\alpha \pi_A + \beta \pi_B}{\alpha + \beta}.$$

At this minimizer one has the identity

$$\min_p L_{\mathrm{out}}(p) \;=\; \alpha\, h(\pi_A) \;+\; \beta\, h(\pi_B) \;+\; \alpha\,\mathrm{KL}(\pi_A \| \bar{\pi}) \;+\; \beta\,\mathrm{KL}(\pi_B \| \bar{\pi}) \;=\; (\alpha + \beta)\, h(\bar{\pi}),$$

where the last equality follows from Lemma 2

$$\alpha\,\mathrm{KL}(\pi_A \| \bar{\pi}) + \beta\,\mathrm{KL}(\pi_B \| \bar{\pi}) \;=\; (\alpha + \beta)\, h(\bar{\pi}) - \alpha h(\pi_A) - \beta h(\pi_B).$$

In contrast, an input–aware $f \in \mathsf{H}_{\mathrm{in}}$ may choose *distinct* probabilities on $A$ and $B$, yielding minimum contribution

$$\min_{f \in \mathsf{H}_{\mathrm{in}}} \mathbb{E}\big[\ell(f(X), Y)\, \mathbf{1}\{X \in A \cup B\}\big] \;=\; \alpha\, h(\pi_A) \;+\; \beta\, h(\pi_B),$$

by predicting $p_A = \pi_A$ on $A$ and $p_B = \pi_B$ on $B$.

Let $f^\star_{\mathrm{out}} \in \mathsf{H}_{\mathrm{out}}$ be a global risk minimizer. Define $\tilde{f} \in \mathsf{H}_{\mathrm{in}}$ to coincide with the input–optimal choices on $A \cup B$ and to agree with $f^\star_{\mathrm{out}}$ on the complement $(A \cup B)^{\complement}$ (eg, just copy $f^\star_{\mathrm{out}}$). This is admissible because $\mathsf{H}_{\mathrm{out}} \subseteq \mathsf{H}_{\mathrm{in}}$. Then

$$\mathbb{E}\big[\ell(f^\star_{\mathrm{out}}(X), Y)\big] - \mathbb{E}\big[\ell(\tilde{f}(X), Y)\big] \;=\; \underbrace{\min_p L_{\mathrm{out}}(p)}_{=(\alpha + \beta)h(\bar{\pi})} - \underbrace{\big(\alpha h(\pi_A) + \beta h(\pi_B)\big)}_{\text{input–aware best on } A \cup B} \;=\; \alpha\,\mathrm{KL}(\pi_A \| \bar{\pi}) + \beta\,\mathrm{KL}(\pi_B \| \bar{\pi}).$$

The last equality follows from Lemma 2 again. Since $\pi_A \neq \pi_B$ and $\alpha, \beta > 0$, their weighted average lies strictly between them:

$$\bar{\pi} \;\in\; \big(\,\min\{\pi_A, \pi_B\},\; \max\{\pi_A, \pi_B\}\big).$$

For Bernoulli distributions, $\mathrm{KL}(\pi \| q) = 0$ if and only if $\pi = q$. Hence

$$\mathrm{KL}(\pi_A \| \bar{\pi}) > 0, \qquad \mathrm{KL}(\pi_B \| \bar{\pi}) > 0.$$

Therefore, the gap between the output-only optimum and the input-aware optimum on $A \cup B$ is strictly positive.

Turning to global risks, by definition

$$\inf_{f \in \mathsf{H}_{\mathrm{out}}} \mathbb{E}[\ell(f(X), Y)] = \mathbb{E}[\ell(f^\star_{\mathrm{out}}(X), Y)].$$

By construction of $\tilde{f}$,

$$\mathbb{E}[\ell(f_{\text{out}}^{\star}(X),Y)] \;\geq\; \mathbb{E}[\ell(\tilde{f}(X),Y)],$$

and since $\tilde{f}$ is only one candidate in $\mathsf{H}_{\text{in}}$,

$$\mathbb{E}[\ell(\tilde{f}(X),Y)] \;\geq\; \inf_{f\in\mathsf{H}_{\text{in}}}\mathbb{E}[\ell(f(X),Y)].$$

Combining these inequalities with the KL identity yields

$$\inf_{f\in\mathsf{H}_{\text{out}}}\mathbb{E}[\ell(f(X),Y)] - \inf_{f\in\mathsf{H}_{\text{in}}}\mathbb{E}[\ell(f(X),Y)] \;\geq\; \alpha\,\mathrm{KL}(\pi_A\|\bar{\pi}) \,+\, \beta\,\mathrm{KL}(\pi_B\|\bar{\pi}) \;>\; 0.$$

This establishes the strict inequality between the optimal risks of the two hypothesis classes and completes the proof.

$\square$

## A.6 ALGORITHMS

## A.7 COMPLEXITY ANALYSIS OF HYPERSYN

In this section, we compare the computational complexity of HyperSyn to traditional output-level fusion methods (e.g., FoE, MoE, nonlinear fusion). We focus on additional overhead on top of querying the black-box experts, since in many realistic settings (e.g., remote API calls) the cost of running the experts dominates.

**Notation.** Let

- $K$ be the number of experts,
- $d_U$ be the dimension of the concatenated expert outputs $U(x) \in \mathbb{R}^{d_U}$,
- $d_x$ be the input dimension,
- $h_F$ be the hidden size of a standard fusion network $F$,
- $h_H$ be the hidden size of the hypernetwork $H$,
- $h_T$ be the hidden size of the target encoder $T_{\theta_x}$,
- $d_v$ be the latent dimension produced by the encoder,
- $C$ be the number of output classes.

For clarity we assume single-hidden-layer MLPs for all networks (same as our implementation); the analysis extends straightforwardly to deeper architectures.

### A.7.1 PARAMETER COUNT

**Traditional fusion methods.** A standard fusion network $F$ taking $U(x)$ as input and producing a $C$-dimensional output via one hidden layer of size $h_F$ has:

$$P_F = \underbrace{d_U \cdot h_F}_{\text{input}\to\text{hidden weights}} + \underbrace{h_F}_{\text{hidden bias}} + \underbrace{h_F \cdot C}_{\text{hidden}\to\text{output weights}} + \underbrace{C}_{\text{output bias}}$$

$$= d_U h_F + h_F + h_F C + C.$$

If we denote the final linear classifier as $g_F$, this parameterization includes both the fusion mapping and the classifier head; we treat FoE/MoE and other fusion variants as having comparable order of parameters in practice.

**HyperSyn.** HyperSyn consists of three components:

1. **Hypernetwork** $H : \mathbb{R}^{d_U} \to \mathbb{R}^{P_T}$, with one hidden layer of size $h_H$, mapping expert outputs $U(x)$ to encoder parameters $\theta_x$. Here $P_T$ is the number of parameters of the encoder $T_{\theta_x}$.

$$P_H = d_U \cdot h_H + h_H + h_H \cdot P_T + P_T.$$

---

**Algorithm 1** HYPERSYN Training (minibatch SGD with AdamW)

---

1: **Inputs:** reference data $\mathcal{D}_{\text{ref}} = \{(x_j, y_j)\}_{j=1}^n$, frozen experts $\{m_i\}_{i=1}^K$ (black-box), loss $\ell$, epochs $E$, batch size $B$, step size $\eta$.

2: **Hyperparameters:** $\lambda_{\text{reg}}$ (oracle-regret), $\lambda_{\text{dist}}$ (distillation), $\rho$ (parameter L2), $\epsilon$ (stability in $w(x)$).

3: **Shapes:** $x \in \mathbb{R}^d$, $U(x) \in \mathbb{R}^{d_u}$ with $d_u = K \cdot C$ (flattened), $v(x), e(x) \in \mathbb{R}^h$, logits in $\mathbb{R}$ (binary) or $\mathbb{R}^C$ (multiclass).

4: Initialize parameters of hypernetwork $H$, fusion encoder $E$, shared predictor $g$. {Weights learned; experts $\{m_i\}$ are frozen and black-box}

5: **for** $t = 1$ to $E$ **do**

6:     **for** minibatch $\mathcal{B} = \{(x, y)\}_{b=1}^B \subset \mathcal{D}_{\text{ref}}$ **do**

7:         **Get expert outputs:** $U(x) = (m_1(x), \ldots, m_K(x))$ for each $(x, y) \in \mathcal{B}$; flatten to $U(x) \in \mathbb{R}^{d_u}$.

8:         **Hypernetwork:** $\theta_x \leftarrow H(U(x)) \in \mathbb{R}^{d_\theta}$ for each $x \in \mathcal{B}$.

9:         **Target encoder:** $v(x) \leftarrow T_{\theta_x}(x) \in \mathbb{R}^h$ for each $x \in \mathcal{B}$.

10:       **Fusion encoder:** $e(x) \leftarrow E(U(x)) \in \mathbb{R}^h$ for each $x \in \mathcal{B}$.

11:       **Predictions:** $\hat{y}(x) \leftarrow g(v(x))$,   $\hat{y}^{\text{u}}(x) \leftarrow g(e(x))$.

12:       **Supervised losses:**

$$\ell_{\text{sup}} \leftarrow \frac{1}{B} \sum_{(x,y) \in \mathcal{B}} \ell(\hat{y}(x), y), \qquad \ell_{\text{aux}} \leftarrow \frac{1}{B} \sum_{(x,y) \in \mathcal{B}} \ell(\hat{y}^{\text{u}}(x), y).$$

13:       **Oracle loss and regret:**

$$\ell^\star(x, y) = \min_{i \in [K]} \ell(m_i(x), y), \quad \ell_{\text{regret}} \leftarrow \frac{1}{B} \sum_{(x,y) \in \mathcal{B}} \max\{\ell(\hat{y}(x), y) - \ell^\star(x, y), 0\}.$$

14:       **Parameter L2:**  $\ell_\theta \leftarrow \frac{1}{B} \sum_{x \in \mathcal{B}} \|\theta_x\|_2^2$.

15:       **Weighted distillation:** compute per-example weights

$$w(x) = \frac{1}{\ell(\hat{y}^{\text{u}}(x), y) + \epsilon} \bigg/ \frac{1}{B} \sum_{(x', y') \in \mathcal{B}} \frac{1}{\ell(\hat{y}^{\text{u}}(x'), y') + \epsilon},$$

        then

$$\ell_{\text{dist}} \leftarrow \frac{1}{B} \sum_{x \in \mathcal{B}} w(x) \|v(x) - e(x)\|_2^2.$$

16:       **Total loss:**

$$\mathcal{L} \leftarrow \ell_{\text{sup}} + \ell_{\text{aux}} + \lambda_{\text{reg}} \, \ell_{\text{regret}} + \lambda_{\text{dist}} \, \ell_{\text{dist}} + \rho \, \ell_\theta.$$

17:       Backpropagate $\nabla \mathcal{L}$; apply AdamW step with learning rate $\eta$ and gradient-norm clipping.

18:     **end for**

19: **end for**

20: **Output:** trained $H$, $E$, and $g$.

---

**Algorithm 2** HYPERSYN Inference

---

1: **Inputs:** test input $x \in \mathbb{R}^d$, frozen experts $\{m_i\}_{i=1}^K$, trained $H, E, g$.

2: Obtain expert outputs and flatten: $U(x) = (m_1(x), \ldots, m_K(x)) \in \mathbb{R}^{d_u}$.

3: Emit per-example parameters: $\theta_x \leftarrow H(U(x))$.

4: Encode input with emitted parameters: $v(x) \leftarrow T_{\theta_x}(x)$.

5: Predict logits: $\hat{y}(x) \leftarrow g(v(x))$.

6: **Return probabilities:**
- Binary: $p(y{=}1 \mid x) = \sigma(\hat{y}(x))$.
- Multiclass: $p(\cdot \mid x) = \text{softmax}(\hat{y}(x))$.

---

Table 4: Leading-order parameter counts (ignoring biases).

| Method | Parameters (leading order) |
|---|---|
| Fusion (FoE/MoE) | $P_F \approx d_U h_F + h_F C$ |
| HyperSyn | $P_{\text{HyperSyn}} \approx d_U h_H + h_H P_T + d_v C$ |

2. **Target encoder** $T_{\theta_x} : \mathbb{R}^{d_x} \to \mathbb{R}^{d_v}$, instantiated as a single-hidden-layer MLP with hidden size $h_T$. For a fixed parameter vector $\theta$, the encoder has

$$P_T^{(\text{struct})} = d_x \cdot h_T + h_T + h_T \cdot d_v + d_v.$$

These are *generated* per instance, but their structural count is independent of the number of instances.

3. **Shared predictor** $g : \mathbb{R}^{d_v} \to \mathbb{R}^C$, which is a fixed (non-generated) linear classifier:

$$P_g = d_v \cdot C + C.$$

The total number of *trainable* parameters in HyperSyn is therefore

$$P_{\text{HyperSyn}} = P_H + P_g,$$

since the encoder parameters are generated on-the-fly and do not introduce additional trainable degrees of freedom beyond the hypernetwork.

For typical tabular settings (moderate $d_x$, small $d_U$, small $h_H$, $h_T$, and $d_v$), we choose $P_T$ and $h_H$ such that $P_{\text{HyperSyn}}$ is of the same order as $P_F$. In particular, in our experiments we use single-layer MLPs with small hidden sizes for both $H$ and $T_{\theta_x}$, which keeps the parameter count comparable to standard fusion architectures.

Table 4 summarizes the leading-order parameter counts ignoring biases. Thus, while HyperSyn introduces a hypernetwork and encoder, careful choice of $h_H, h_T, d_v$ keeps the total number of trainable parameters within the same order as traditional fusion methods.

### A.7.2 INFERENCE COST

We now compare the per-instance computational cost for inference, excluding the cost of querying the experts themselves. Let $\text{FLOPs}(\cdot)$ denote the number of floating point operations for a forward pass.

**Traditional fusion.** A one-hidden-layer fusion network $F$ followed by a classifier head requires:

$$\text{FLOPs}_{\text{fus}}(x) \approx \text{FLOPs}\big(F(U(x))\big) = \mathcal{O}(d_U h_F + h_F C).$$

This is a single forward pass through an MLP of size $(d_U, h_F, C)$.

**HyperSyn.** HyperSyn performs three forward passes per instance:

1. Hypernetwork: $H(U(x))$ to obtain $\theta_x$,
2. Encoder: $T_{\theta_x}(x)$ to obtain latent $v(x)$,
3. Predictor: $g(v(x))$ to obtain logits.

The corresponding cost is

$$\text{FLOPs}_{\text{HyperSyn}}(x) \approx \underbrace{\mathcal{O}(d_U h_H + h_H P_T)}_{\text{hypernetwork}} + \underbrace{\mathcal{O}(d_x h_T + h_T d_v)}_{\text{encoder}} + \underbrace{\mathcal{O}(d_v C)}_{\text{predictor}}.$$

Since both $H$ and $T_{\theta_x}$ are single-layer MLPs with modest hidden sizes, the additional cost compared to fusion is roughly on the order of one extra small MLP per instance.

**Practical overhead.** For a single-example forward pass, the additional cost of HyperSyn is small. However, hypernetworks require generating a distinct parameter vector for each instance, which makes naive batch computation less straightforward and can reduce parallelism. One implementation strategy is to materialize all generated parameters for a batch and then apply the corresponding encoders in parallel, at the cost of increased memory usage.

Nevertheless, in many realistic black-box settings, the dominant cost is the evaluation of large experts (e.g., remote classifier APIs, proprietary models). Let $\text{FLOPs}_{\text{experts}}(x)$ denote the total cost of querying the experts. Then the relative overhead satisfies

$$\frac{\text{FLOPs}_{\text{HyperSyn}}(x) - \text{FLOPs}_{\text{fus}}(x)}{\text{FLOPs}_{\text{experts}}(x)} \ll 1$$

whenever the experts are substantially larger than the small MLPs used in HyperSyn.

## A.8 PIECEWISE XOR (PXOR) DATASET

To further evaluate expert fusion under heterogeneous local rules, we construct the *Piecewise XOR (PXOR)* dataset, a variant of the classic XOR problem. Each data point is a triplet $x = (x_1, x_2, x_3)$ sampled uniformly from $[-1, 1]^3$. The label is determined by a piecewise rule:

$$y = \begin{cases} (x_1 > 0) \vee (x_2 > 0), & \text{if } x_3 > 0 \quad \text{(OR rule)} \\ (x_1 \cdot x_2 < 0), & \text{if } x_3 \leq 0 \quad \text{(XOR rule)}. \end{cases}$$

Thus, half the space ($x_3 > 0$) is governed by an OR classifier, while the other half ($x_3 \leq 0$) is governed by XOR. By toggling the piecewise switch ($x_3$), PXOR can be "recovered" back to a standard XOR problem, making it a natural extension that mixes domains with different local labeling functions.

We also define simple black-box experts that operate on $x_1$ and $x_2$ individually, producing high probabilities when $x_1 > 0$ or $x_2 > 0$. These experts perform reasonably on OR regions but fail to capture XOR structure. Consequently, PXOR stresses the limitations of output-only fusion methods: a single expert is informative in the OR region and the fusion of expert outputs is informative in the XOR region, but confusion will happen without leveraging the raw input.

Figures 3, 4, 5, and 6 visualizes the classification accuracy of the different methods on the PXOR dataset, which is partitioned into regions governed by OR and XOR rules. Among all methods, only HyperSyn achieves correct classification across all regions.

In expert-covered regions such as Q2 in the OR rule area (where $x_3 > 0$), all methods attain high accuracy. In fusion-covered regions such as Q4 in the XOR area, MoE fails to classify points correctly, whereas methods that perform fusion of experts (FoE, FoE+X, and HyperSyn) are able to capture the correct quadrant structure.

In contrast, residual regions such as Q1 in the XOR area (where two positive inputs map to label 0) can only be modeled by FoE+X and HyperSyn, since they read the additional feature $x_3$. However, FoE+X attains lower accuracy and fails on some quadrants, while HyperSyn achieves consistently high accuracy across all quadrants, benefiting from its more expressive architecture under a comparable parameter budget.

## A.9 DETAILS OF SYNTHETIC DATA: MULTI–DOMAIN MULTI–FUNCTION (MDMF)

**Motivation.** To probe the limits of black–box expert fusion and instance–wise synthesis, we design a controllable synthetic benchmark that exhibits both (i) *domain heterogeneity* (covariate shift across $K$ domains) and (ii) *function heterogeneity* (each domain uses its own labeling function). This produces a piecewise linear decision rule that no single global linear model can fit and that purely output–space fusion may fail to reconstruct, while instance–wise synthesis can adapt by emitting input–conditioned parameters.

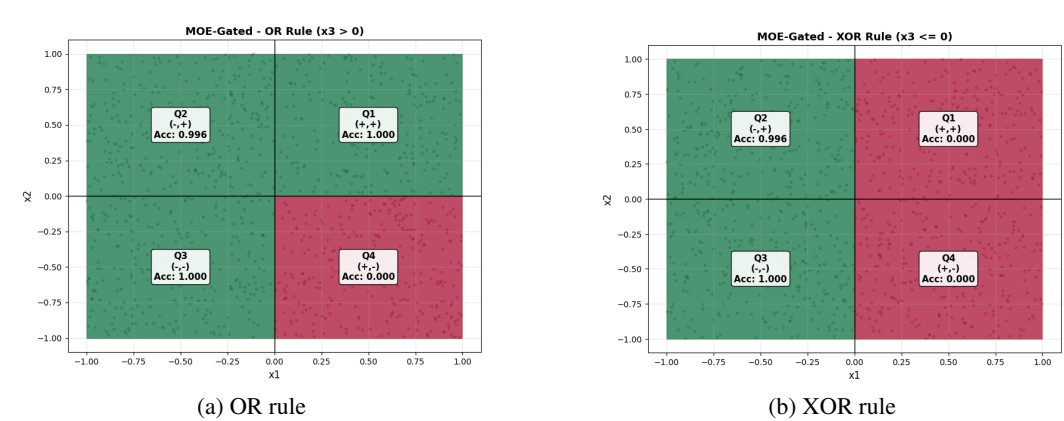

(a) OR rule

(b) XOR rule

Figure 3: Accuracy of PXOR Dataset by MOE-Gated

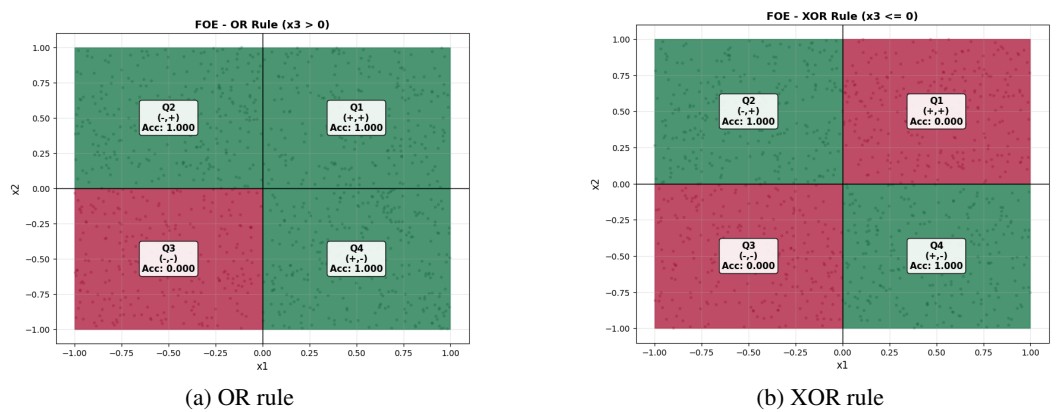

(a) OR rule

(b) XOR rule

Figure 4: Accuracy of PXOR Dataset by FOE

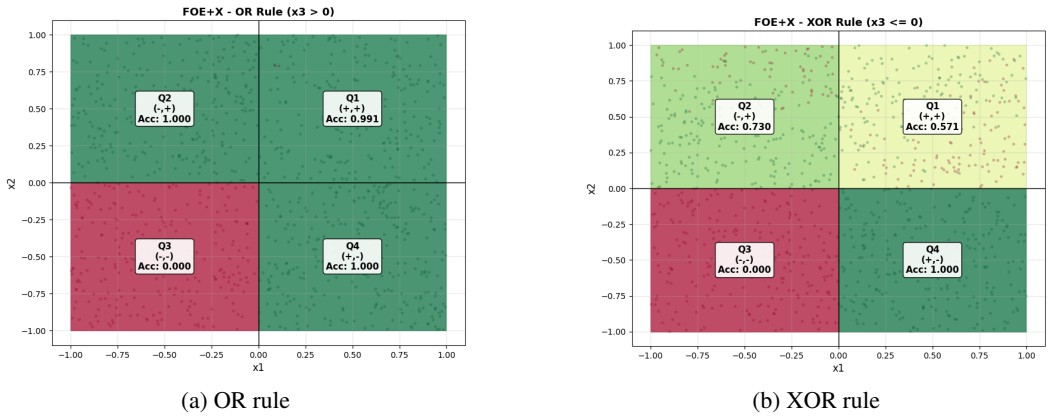

(a) OR rule

(b) XOR rule

Figure 5: Accuracy of PXOR Dataset by FOE+X

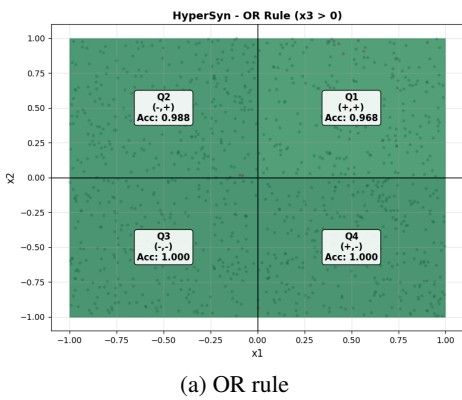 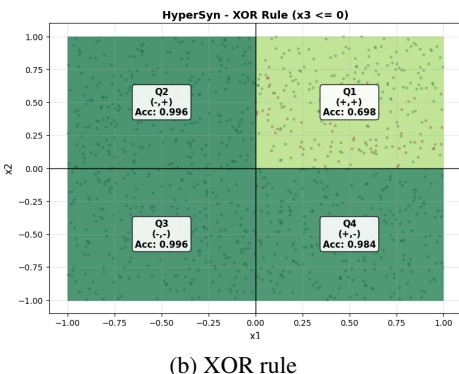

(a) OR rule    (b) XOR rule

Figure 6: Accuracy of PXOR Dataset by HyperSyn

### A.9.1 GENERATIVE PROCESS

Let $K$ denote the number of domains and $d$ the input dimension. For each domain $k \in [K]$, we sample a *domain center* $\mu_k \in \mathbb{R}^d$ and a *domain–specific weight vector* $w_k \in \mathbb{R}^d$:

$$\mu_k \sim \mathcal{N}(0, \sigma_\mu^2 I_d), \qquad w_k \sim \mathcal{N}(0, \sigma_w^2 I_d),$$

with $\sigma_\mu$ (`center_scale`) and $\sigma_w$ (`weight_scale`) as controls.

A labeled example $(x, y)$ is generated by:

$$k \sim \text{Unif}\{1, \ldots, K\}, \quad x \mid k \sim \mathcal{N}(\mu_k, \sigma_x^2 I_d), \quad y = \mathbb{1}\{w_k^\top x > 0\},$$

where $\sigma_x$ governs within–domain spread. Thus each domain induces a linear Bayes classifier $x \mapsto \text{sign}(w_k^\top x)$ around its own center $\mu_k$. The overall task is *binary* ($C = 2$), but the global decision boundary is a union of $K$ linear pieces, one per domain.

### A.9.2 CONTROLLABLE PARAMETERS

- **Center Scale/Domain separation** ($\sigma_\mu$): larger values increase inter–domain covariate shift; small values make domains overlap.
- **Within–domain spread** ($\sigma_x$): larger spread blurs the local margin; smaller spread yields cleaner per–domain linearity.
- **Weight Scale/Function heterogeneity** ($\sigma_w$): larger values produce diverse $w_k$ leading to strongly different local boundaries; smaller values align domains and ease global fitting.
- **Dimensionality** ($d$): higher $d$ induces higher–rank interactions and more room for domain–specific projections.
- **Number of domains** ($K$): increases the number of pieces in the global rule and the difficulty of selection/gating.

A convenient signal–to–noise proxy is $\text{SNR}_{\text{dom}} := \sigma_\mu / \sigma_x$: high $\text{SNR}_{\text{dom}}$ yields well–separated domains (easy to route), whereas low $\text{SNR}_{\text{dom}}$ forces methods to rely on function cues rather than covariates alone.

### A.9.3 BLACK–BOX EXPERTS AND SIGNALS

To instantiate black–box signals $U(x)$ for methods that use expert outputs, we adopt the following simple protocol (not required by the generator but natural for evaluation):

(a) **Per–domain experts.** Train $K$ frozen linear experts $\{m_i\}_{i=1}^K$ (e.g., logistic regression) on the *training subset* from domain $i$ only, then freeze them. At test time, each $m_i$ outputs a probability $m_i(x) \in [0, 1]$ for $y=1$.[1]

---

[1] Any black–box classifier may be used; only outputs are consumed.

(b) **Signals.** For binary MDMF, the concatenated signals are $U(x) \in \mathbb{R}^K$ (one probability per expert). For multiclass variants (if extended), $U(x) \in \mathbb{R}^{K \cdot C}$.

This setting reflects realistic constraints: experts are trained elsewhere on different domains and expose *outputs only*.

### A.9.4 How MDMF Stresses Methods (Link to Taxonomy)

MDMF naturally induces all three regions in §3.1:

- **Expert–covered** $\Omega_{\text{exp}}$: near $\mu_k$ with large margin $|w_k^\top x|$, expert $m_k$ is near–optimal; selection achieves zero regret.
- **Fusion–covered** $\Omega$: in overlapping areas where no single expert suffices but complementary outputs across $\{m_i\}$ carry enough information, a nonlinear fusion on $U(x)$ can recover $y$.
- **Residual** $\Omega$: when domains overlap and the $w_k$ disagree sharply, $U(x)$ alone can be ambiguous (experts confident yet contradictory). Output–only functions $g(U(x))$ are information–limited; an input–aware synthesizer must leverage $x$ to form the correct projection.

The global rule is a union of $K$ per–domain linear separators. A *fixed* fusion $g(U(x))$ cannot in general implement the correct $x \mapsto \text{sign}(w_k^\top x)$ mapping when the relevant separator changes with $k$ and $U(x)$ is ambiguous. HYPERSYN instead conditions on $U(x)$ to emit per–example parameters $\theta_x$ and then applies $T_{\theta_x}$ to *the input* $x$, effectively synthesizing a local projection aligned with the operative $w_k$ (or its interpolation). This is precisely the kind of residual regime where instance–wise synthesis can surpass both selection and output–space fusion.

### A.9.5 Practical Defaults and Reproducibility

Unless otherwise noted, we use $K = 30$, $d = 100$, `center_scale` $= 3.0$, `weight_scale` $= 3.0$ and balanced splits ($n_{\text{train}} = 2000$, $n_{\text{val}} = 1000$, $n_{\text{test}} = 1000$), and 80% of domains is used for training.

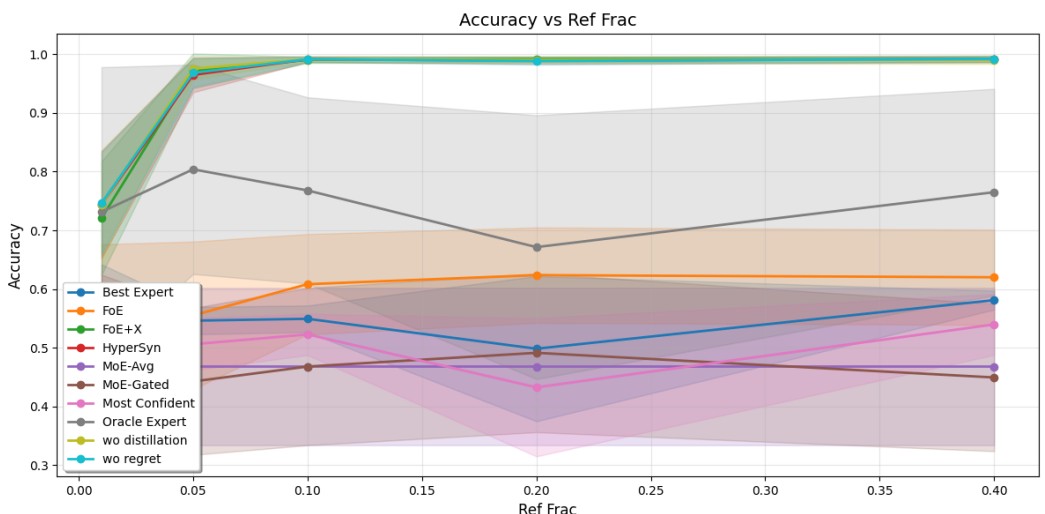

Figure 7: reference data fraction versus accuracy for MDMF data

### A.10 Details of Baselines

- **Oracle Expert** Where the expert with the best accuracy for the data points is selected as output. It is treated as the ideal but not realistic baseline as we assume the data labels are available in the test data to select the oracle.
- **Best Expert** where the best expert is selected using training data. And the single best model is used to evaluate the test data.

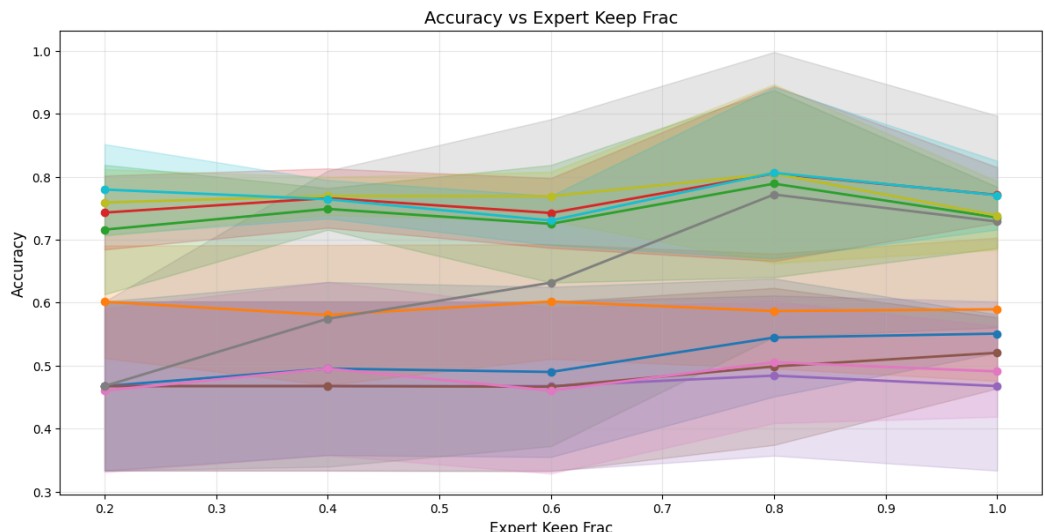

Figure 8: Expert keeps fraction versus accuracy for MDMF data

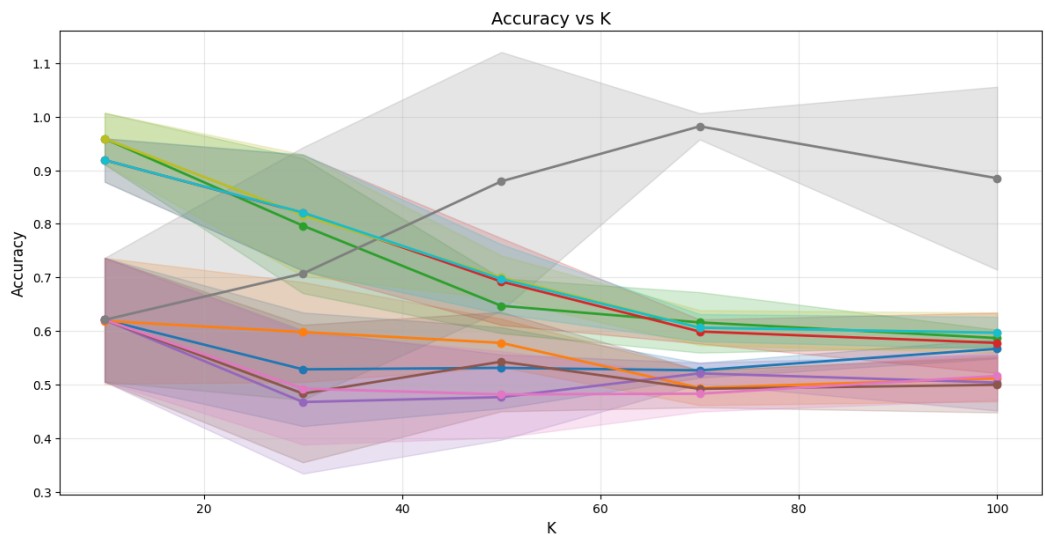

Figure 9: Number of domains versus accuracy for MDMF data

- **Most Confidence(Pearce et al., 2021; Chen et al., 2023)** where the method selects the expert that outputs the largest probability for a class.
- **MoE-AVG** A traditional ensembling technique where we simply take the average of the outputs of experts as the prediction.
- **MoE-Gated(Jacobs et al., 1991; Jordan & Jacobs, 1994; Masoudnia & Ebrahimpour, 2014)** where there is a gating network that maps the input to weight coefficients to combine different experts.
- **FoE(Wang et al., 2024)** which fuses outputs of expert models with complementary knowledge of the data distribution.
- **FoE+X** An extension of FoE with concatenated x and outputs of experts as input of fusion model.

## A.11 DETAILS OF REAL WORLD DATA

Table 5 shows the statistics of real world datasets used in the paper.

Table 5: Summary of real world benchmark datasets.

| Dataset | #Samples | #Features | #Classes | #Domains | Type |
|---|---|---|---|---|---|
| Adult | 48,842 | 15 | 2 | 5 | Human |
| Mushroom | 8,124 | 23 | 2 | 7 | Biology |
| Gesture | 9,873 | 32 | 5 | 7 | Gesture phase |
| Mice Protein | 1,080 | 76 | 8 | 72 | Genomic |
| HAR | 10,299 | 561 | 6 | 72 | Human activity |
| Isolet | 7,797 | 617 | 26 | 150 | Speech |
| Eucalyptus | 736 | 14 | 5 | 16 | Biology |
| Vowel | 990 | 10 | 11 | 15 | Speech |
| Gas Drift | 13,910 | 128 | 6 | 10 | Chemistry |

**adult(Kohavi, 1996)** The prediction task is to determine whether a person makes over 50K a year. Extraction was done by Barry Becker from the 1994 Census database. The 'race' is used as the domain.

**mushroom(Lincoff, 1981)** This dataset includes physical characteristics of 23 species of gilled mushrooms in the Agaricus and Lepiota families. They are classified into poisonous or edible. The 'habitat' is used as the domain.

**gesture(Madeo & Peres, 2013)** The dataset is composed of features extracted from 7 videos with 3 people gesticulating, aiming at studying Gesture Phase Segmentation. The different videos are used as the domains. Notice that multiple videos could belong to the same person.

**mice_protein(Higuera et al., 2015)** It is a gene expression data measured in the cerebral cortex of 8 classes of control and Down syndrome mice exposed to context fear conditioning, a task used to assess associative learning. We apply K-means clustering using 72 clusters(as there are a total of 72 mice) to obtain the domain labels.

**har(Anguita et al., 2013)** It is a dataset built from the recordings of 30 subjects performing activities of daily living (ADL) while carrying a waist-mounted smartphone with embedded inertial sensors. The task is to predict six activities performed by subjects(eg,SITTING, STANDING, LAYING). We apply K-means clustering using 30 clusters(as there are a total of 30 subjects) to obtain the domain labels.

**isolet(Cole & Fanty, 1991)** ISOLET (Isolated Letter Speech Recognition) is a speech data set that contains 150 subjects speaking the name of each letter of the alphabet twice. The task is to predict the letter. We apply K-means clustering using 150 clusters(as there are a total of 150 subjects) to obtain the domain labels.

**eucalyptus(Bulluch (1992))** It is a dataset to predict the Utility of eucalyptus. We use 'Abbrev' - site abbreviation as the domains.

**vowel(Deterding & Robinson, 1988)** It is a Vowel Recognition data set that predicts eleven steady state vowels of British English using a specified training set of lpc derived log area ratios. We apply K-means clustering using 15 clusters(as there are a total of 15 speakers) to obtain the domain labels.

**gas_drift(Vergara, 2012)** The dataset contains measurements from 16 chemical sensors utilized in simulations for drift compensation in a discrimination task of 6 gases at various levels of concentrations, including (1: Ethanol; 2: Ethylene; 3:Ammonia; 4: Acetaldehyde; 5: Acetone; 6: Toluene). The data is organized into ten batches, each containing the number of measurements per class and month. For example, Batch 1 is collected in Months 1 and 2, and Batch 7 is collected in Month 21. We use the batch as domains.

Table 6: Fraction of test points on which HyperSyn's prediction loss is lower than the oracle expert, and the corresponding average oracle regret. We report the mean $\pm$ standard deviation over 5 runs.

| Dataset | Fraction | Average Regret |
|---|---|---|
| Adult | 0.0917±0.0038 | 0.0815±0.0033 |
| Mushroom | 0.0612±0.0438 | 0.1376±0.0818 |
| Gesture | 0.1386±0.0139 | 0.1021±0.0754 |
| Mice protein | 0.0815±0.0055 | 0.1295±0.0323 |
| HAR | 0.1431±0.0608 | 0.0319±0.0048 |
| Eucalyptus | 0.2878±0.0541 | 0.4190±0.1584 |
| Gas Drift | 0.1130±0.0647 | 0.0898±0.0533 |
| Vowel | 0.2424±0.0479 | 0.1748±0.0549 |
| Isolet | 0.1575±0.0391 | 0.0092±0.0024 |

## A.12 FRACTION OF TEST POINTS WHERE HYPERSYN EXCEEDS THE ORACLE

Table 6 reports, for each dataset, the fraction of test points on which HyperSyn's loss exceeds the oracle expert's loss, together with the average oracle regret. We show the mean and standard deviation over 5 independent runs.

Overall, even without knowing the true label and oracle, HyperSyn wins the oracle expert on a small fraction of test instances on most datasets, and the average regret is close to zero for several datasets. This indicates that HyperSyn tracks the oracle expert closely in practice, while still providing the benefits discussed in the main text under distribution shift.

