# OpenReview forum: "HyperSyn: Synthesizing Instance-wise Model by Fusing Blackbox Expert via Hypernetwork"
_ICLR.cc/2026/Conference — ICLR 2026 Conference Withdrawn Submission_

### Official Review · Reviewer_Ebmk · 2025-10-15

**Soundness:** 3
**Presentation:** 2
**Contribution:** 3
**Rating:** 4
**Confidence:** 3

**Summary:**

This paper addresses the problem of *black-box expert synthesis*, where multiple pretrained models (experts) are only accessible through their output logits or probabilities. The authors identify that existing output-level fusion strategies (e.g., expert selection, mixture-of-experts, nonlinear aggregation) cannot handle cases where all experts fail or disagree, because the necessary information is absent from their outputs.

To formalize this, the paper introduces a three-region taxonomy (expert-covered, fusion-covered, and residual) and shows that output-only fusion has an intrinsic lower bound on achievable risk in the residual region.
To overcome this, the authors propose HyperSyn, a hypernetwork-based synthesis framework that takes expert outputs (U(x)) to generate sample-specific encoder parameters, which are then used to re-encode the original input (x). This design allows the model to revisit input information conditioned on expert predictions, while maintaining an *oracle-dominance* constraint to ensure that the synthesized model does not perform worse than the best expert on average.

Overall, the paper provides an interesting conceptual bridge between “output fusion” and “input-conditioned representation synthesis,” offering theoretical arguments and empirical results on benchmark datasets.

**Strengths:**

* **Novel conceptual insight:** The paper presents a clear and compelling perspective on the *information boundary* of black-box fusion, formalizing when and why output-only aggregation fails. The proposed three-region taxonomy (expert/fusion/residual) is conceptually elegant and could influence future research on ensemble learning and model combination.
* **Methodological originality:** The use of a hypernetwork to generate per-instance encoder parameters conditioned on expert outputs is an interesting and underexplored direction for black-box expert synthesis.
* **Safety guarantee:** The oracle-regret constraint is a well-motivated design to ensure “safe improvement,” which is practically meaningful in safety-critical settings such as medical or industrial applications.
* **Theoretical framing:** Theoretical results provide intuition about representational limits of output-only models and motivate why revisiting the input can help.

**Weaknesses:**

* **Limited experimental coverage:** The experiments are mostly conducted on smaller-scale classification tasks with relatively simple encoders. The paper lacks evaluation on larger and more diverse architectures (e.g., ViT, small LLMs) that better reflect the computational and representational complexity of modern expert systems.
* **Scalability and cost:** Hypernetworks typically scale poorly with model size, since they must output full encoder parameters for each instance. The paper does not sufficiently discuss or measure the computational overhead or memory footprint, which raises doubts about practical scalability.
* **Marginal gains over baselines:** The empirical improvement over strong fusion baselines such as FoE and FoE+X appears modest and sometimes within standard deviation. The results do not yet convincingly demonstrate that the proposed method’s additional complexity yields consistent benefits.

**Questions:**

1. How does HyperSyn scale when the base encoder is large or complex?
2. The improvement over FoE and FoE+X baselines is modest. Can you analyze in which regimes (e.g., OOD degree, expert disagreement) HyperSyn provides the largest gains?
3. Could HyperSyn be applied to generative or multimodal experts (e.g., text-to-image, speech-to-text), and what would need to change for such settings?
4. What is the actual runtime and parameter overhead of training the hypernetwork compared to simple output fusion?

*(If the authors can provide convincing large-scale results or clear cost-performance analyses, my recommendation could increase.)*

---

> ### Author Response · Authors · 2025-11-19
>
> We thank the reviewer for the constructive feedback. Below, we address the weaknesses and questions in detail.
>
> 1. Limited experimental coverage
>
> Our goal is to isolate and study the black-box expert synthesis problem in a controlled tabular setting with explicit distribution shifts and well-understood baselines. This is why we deliberately chose datasets and encoders where we can simulate and measure the residual region.
>
> We agree that confirming HyperSyn’s effectiveness and practicality on more complex architectures would strengthen the paper. Conceptually, HyperSyn is orthogonal to the base architecture: the hypernetwork only needs to generate some encoder parameters conditioned on U(x), and the shared predictor can be any differentiable head.
>
> In the revision, we will explicitly state this scope limitation and outline how HyperSyn could be instantiated on larger models.
>
> 2. Scalability and cost.
>
> HyperSyn is designed to mitigate the usual hypernetwork scaling problem in two ways:
>
> - Only the encoder is generated; the predictor is shared. We do not generate full model parameters for each instance.
> - Low-dimensional conditioning and small MLPs. The conditioning input U(x)  is the concatenated expert logits/probabilities, which are typically low-dimensional compared to raw inputs.
>
> From a complexity standpoint, a standard fusion model computes y_fus  (x)=F(U(x))— one forward pass through F. And HyperSyn computes one forward pass through the hypernetwork H(U(x))  to obtain θ_x, then passes through the encoder T_(θ_x ) (x), then passes through the shared predictor.
>
> Thus, inference cost per instance is roughly “fusion + one small extra MLP”. For large experts served as black-box APIs (e.g., remote models), the dominant cost is typically in calling the experts themselves.
>
>
>
>
> 3. Marginal gains over baselines
>
> Our intention is not to claim large numerical improvements everywhere, but rather to show that HyperSyn does at least as well as strong fusion baselines in “easy” regimes, and to demonstrate that it provides robust advantages in precisely those settings where output-only fusion is inherently limited.
>
> This is most visible in the PXOR / MDMF synthetic experiments, where we explicitly construct regimes in which experts are individually strong on their domains, and output-only fusion breaks down under cross-domain or compositional shift.
>
>
> For the questions:
>
> 1. How does HyperSyn scale when the base encoder is large or complex?
>
> A naïve hypernetwork that generates a full encoder would indeed be expensive. However, HyperSyn is compatible with several scaling strategies:
>
> - Generate only small adapter layers that plug into a frozen large encoder. The hypernetwork output size then scales with adapter size, not the full model.
>
> - Generate parameters only for a subset of layers (e.g., final few blocks), while sharing the rest across instances.
>
> Formally, we can replace T_(θ_x) by a parametrized adapter in an existing encoder and let θ_(x ) represent only the adapter’s parameters. The architecture and training objectives of HyperSyn remain unchanged.
>
> 2. The improvement over FoE and FoE+X baselines is modest.
>
> Empirically and theoretically, HyperSyn is designed to help most in residual regions, characterized by high expert disagreement or uniformly high expert loss (no expert is clearly good), and distribution shift where the experts and their output-only fusion are systematically biased.
>
>
>
> 3. Could HyperSyn be applied to generative or multimodal experts?
>
>
> Conceptually, yes. HyperSyn only assumes a set of black-box experts that map an input to some output, and access to the raw input for the downstream system.  For generative tasks, one could use HyperSyn to produce conditioning parameters in a generative model. We will clarify this in the discussion section, highlighting HyperSyn as a general pattern for leveraging expert outputs as conditioning signals in multi/expert systems, and sketch how this could be applied in multimodal scenarios.
>
> 4.What is the overhead of training the hypernetwork compared to simple output fusion?
>
>
> HyperSyn adds a small hypernetwork that takes U(x)  as input, and a generator-based encoder T_(θ_x) for x, while sharing the predictor g. In terms of parameters, let P_F  be the parameter count of a typical fusion model F(U(x)), and let P_H be the hypernetwork params, and P_T  the encoder params. Then HyperSyn’s parameters are roughly P_H+P_T+P_g, versus P_F+P_g  for “outputs-only + predictor”.
>
> In terms of runtime, A fusion method does one forward pass through F and g. And HyperSyn adds one forward pass through H and T_(θ_x ), which introduces overhead that makes the batch computation difficult. For large experts served as black-box APIs, the dominant cost is typically in calling the experts themselves; HyperSyn’s additional cost is small relative to that.
>
>
>
> We thank the reviewer again for the concrete suggestions. We believe that the revision will address the main concerns raised.

---

> > ### Comment · Reviewer_Ebmk · 2025-11-21
> > **Thank the authors for the reply.**
> >
> > The authors provide a positive and detailed rebuttal, however, the response does not include new empirical evidence or concrete scalability evaluations, and thus does not fully resolve my concerns regarding practical overhead and marginal gains over baselines. I therefore maintain my original score.

---

> > > ### Author Response · Authors · 2025-11-25
> > >
> > > Thanks for the response. We uploaded the revised paper with changes highlighted in blue. To address your concern, we made following changes:
> > >
> > > 1. We clarified the scope of our experiments in Section 5.1, explicitly stating that we focus on tabular, multi-expert settings with controllable distribution shifts, to better motivate the chosen benchmarks. We also added a short subsection (Sec. 4.5) to emphasize that the HyperSyn architecture can be extended to more complex encoders and larger-scale settings.
> > >
> > > 2. We included a complexity analysis of parameter count and inference cost in Appendix A.7, comparing HyperSyn against traditional fusion methods. This analysis shows that the additional computation introduced by HyperSyn is moderate, and that in realistic black-box scenarios the dominant overhead in practice comes from querying the experts rather than from the small hypernetwork and encoder.
> > >
> > > Regarding empirical performance, the proposed method achieves clear gains over the second-best baseline on four real-world datasets, including:
> > > Eucalyptus (0.5257 vs. 0.4635), Gas Drift (0.8954 vs. 0.8453), Vowel (0.4111 vs. 0.3949), and Isolet (0.8012 vs. 0.7700), with statistical significance, and is marginally better on most of the remaining datasets.
> > >
> > > In particular, we visualize the accuracy of different methods on the PXOR dataset in Appendix A.8. These plots show that HyperSyn attains high accuracy across all regions, whereas the baselines fail in some of the OR/XOR subregions, illustrating that the performance gains can be substantial while incurring only moderate computational overhead.
> > >
> > > We also made other changes for the concerns raised by other reviewers, which you may check in other responses. It would be thankful if you could raise the score if you find the changes satisfactory. Any additional comments are welcome.

---

### Official Review · Reviewer_K3kz · 2025-10-27

**Soundness:** 2
**Presentation:** 1
**Contribution:** 2
**Rating:** 2
**Confidence:** 4

**Summary:**

This submission proposes a novel fusion method for model predictions. The authors consider a regime where several blackbox api experts are called on the same data. Their method uses that signal on two paths, one where a learned fusion directly leads to a prediction. The other uses the expert outputs as conditioning for a hypernetwork that leads to a second prediction. In experimental evaluation, the method appears to outperform other model fusion and ensembling methods methods.

**Strengths:**

- The setting of fusing expert API signals beyond ensembling appears interesting and relevant.
- The decision families are convincing, and I encourage the authors to evaluate based on these categories. Ideally, a system would match or outperform the strongest single actor on each of them.
- The idea of combining learned softmax fusion with a hypernetwork appears novel. There might be something there, to leverage the signal from experts for a hypernetwork.

**Weaknesses:**

- The writing could be improved to emphasize clarity, especially in section 3 and 4. There are entire paragraphs that are repetitions of previous paragraphs. Symbols get defined and re-defined, sometimes differently ($U(x)$ for example). The authors included a proof for a residual region that I think could have just been a definition. There is quite a lot of boilerplate math, e.g. for ERM CE, etc. At the same time or because of that, crucial aspects of the method remain unclear, most notably the meta policy to decide whether to use the hyper network or fusion signal, see below.
- Method remains unclear. The main method identifies decision families (ll 137-147), and proposes a meta policy (ll 156-160) to decide what part of the signal to use. While that seems quite involved, it also doesn't map onto the actually proposed method in section 4. Out of the 4 decision families, only two seem to be represented. What is more, the meta policy to decide between the two predictions that seems to be made for every input seems absent. It remains unclear, what prediction is used for the experiment, and how picking a signal was chosen. There is a chance that the authors use the hypernetwork prediction always - but then what's the point of the rest of the method?
- The experiment section shows a large number of experiments on which the proposed method appears to perform well compared to other fusion methods. Notably, in tables 2 and 3, their proposed method performs worse than the oracle expert - which was the expressed goal to match. Given these results, I strongly miss experiments to build understanding or test assumptions. First, the added regret and distillation loss don't appear to have much of an impact - which should trigger some careful follow up experiments or lead to the decision to not include these components for reduced complexity. Second, The authors introduce the dicision families, but never check whether a) that intuition holds up in practice, or b) whether their hypernetwork based system does indeed help fuse predictions in cases where the oracle predictions lack information.

**Questions:**

- How are predictions chosen between fusion and hypernetworks?
- Why the choice to only feed the expert outputs into the hypernetwork, especially given the classification as not enough signal in expert outputs?
- How did you intend to use the decision families, if not to test your system against them or use them as switch between the different signal paths?

---

> ### Author Response · Authors · 2025-11-19
>
> We thank the reviewer for constructive feedback, and we address the main concerns below and will revise the paper accordingly.
>
> For the weaknesses:
>
> 1. The writing could be improved to emphasize clarity.
>
> We agree that clarity and conciseness can be improved, especially in Sections 3 and 4. Specifically, we will:
>
> - Keep the decision-family taxonomy and three-region picture once, and refer back to it instead of rephrasing it multiple times.
> - Move basic definitions and other details to the appendix (or shorten them), so they don’t obscure the main ideas.
> - Simplify the residual-region formalism. As you suggest, the “residual region” can be stated more cleanly as a definition rather than framed as a heavy theorem. We will streamline this part and move technical proofs to the appendix.
>
> 2. Method remains unclear.
>
>
> In all reported experiments, the final prediction used for evaluation is the input-based branch. The outputs-only branch is never used directly at test time for final predictions. Its roles are:
>
> - Auxiliary supervised head: It is trained to predict the label directly from U(x), providing a strong “fusion-only” baseline inside HyperSyn.
> - Teacher for distillation: It provides the latent e(x)  that the input-aware latent v(x)  is regressed towards when the fusion view is reliable (via reliability-weighted distillation).
>
> We will explicitly state this in the paper.
>
> The decision families and meta policy in Section 3 are an analytical framework, not an explicit gating module we implement. Specifically:
>
> - We define four families. In theory, a meta-policy could choose which family to use per input.
> - In practice, our method instantiates one concrete meta-policy in the class A_syn, with the output-only fusion branch acting as a teacher rather than a separate policy we switch to.
>
> So there is no per-example discrete switch between the two branches at inference time. Instead, the losses implement a soft, loss-driven policy.
>
>
>
>
> 3. Oracle vs. HyperSyn, and impact of regret & distillation terms. And use of decision families.
>
>
> The goal we state is to approximately satisfy oracle dominance while improving performance in regimes where experts/fusion fail. Our theoretical results show that strict oracle dominance plus improvement everywhere is generally impossible in a non-trivial way. In the current draft, some phrasing may have been too strong (e.g., sounding like we always match the oracle). We will refine the language to emphasize that we minimize oracle regret and empirically observe low regret on most datasets, rather than claim perfect matching.
>
>
> On the impact of the regret and distillation losses,  in our internal experiments, we observe that removing the oracle-regret term increases the fraction of points where HyperSyn underperforms the oracle and raises average regret. And removing the distillation term reduces performance, particularly in fusion-covered regimes.
>
> For the weakness of the use of decision families and missing experiments, the synthetic PXOR and MDMF setups are explicitly constructed to create residual regimes. In these regimes, output-only fusion provably cannot recover some patterns, and we indeed observe that HyperSyn maintains performance while MoE/fusion degrades.
>
> For the revision, we will add plots of the input space with points colored by which regime is active.
>
>
> Response to questions:
>
> 1. How are predictions chosen between fusion and hypernetworks?
>
> As clarified above, they are not chosen between. The final prediction used in all experiments is always the input-based branch. The fusion branch is auxiliary, used only for training (supervised loss and distillation) and analysis.
>
> 2. Why the choice to only feed the expert outputs into the hypernetwork?
>
>
> To maintain a clean black-box design, the hypernetwork H(U(x))  sees only expert outputs and acts as a controller that configures how the raw input x should be encoded via T_(θ_x ) (x). We argue that expert outputs are insufficient as the sole input to a predictor in residual regimes, but they are very informative as context for how to read x. The separation H(U(x))vs. T_(θ_x ) (x)  makes this explicit.
>
> 3. How did you intend to use the decision families, if not to test your system against them or use them as switch between the different signal paths?
>
>
> The decision families serve primarily as:
>
> - A conceptual scaffold to understand where classical selection, MoE, fusion, and instance-wise synthesis help or fail; and
> - A basis for our theoretical limitations results (e.g., on what fusion can or cannot do in residual regions).
>
> In practice, we instantiate a specific member of A_syn (HyperSyn) and use the fusion family as an auxiliary teacher, not as a discrete switch.
>
> We thank the reviewer again for the detailed and constructive critiques. We believe that the revision will significantly improve both the presentation and soundness of the paper.

---

> ### Author Response · Authors · 2025-11-25
>
> Thanks for the response. We uploaded the revised paper with changes highlighted in blue. To address your concern, we made following changes:
>
> 1. We removed repeated definitions and tightened the presentation in Section 4. In particular, we avoid redefining U(x) and the model components, and we eliminated boilerplate math such as the cross-entropy definition in Section 4.5.1. The formal definition of the residual set with margin has been moved to the appendix to improve the flow of the main text.
>
> 2. We clarified the role of the two branches by explicitly stating in Section 4.5 that:
>    “At test time, the final prediction is always $\hat{y}(x)$ from the input-aware branch; the outputs-only branch is auxiliary and is used only during training.”
>
> 3. We added visualizations of the accuracy of different methods on the PXOR dataset in Appendix A.8. These plots show that HyperSyn achieves high accuracy across all regions, whereas the baselines fail in some of the OR/XOR subregions.
>
> We also make other changes for the concerns raised by other reviewers which you may check in other responses. It would be thankful if you could raise the score if you find the changes satisfactory. Any additional comments are welcome.

---

> > ### Comment · Reviewer_K3kz · 2025-11-26
> > **Reviewer Response**
> >
> > I thank the authors again for their response and revision - as well as for spotting my comment in another reviewer's thread; apologies for the confusion.
> >
> > I've reviewed the revised version of the paper. It drops a repetitive paragraph and changes individual words to sharpen here and there. While I believe that is a step in the right direction, it does not address the general lack of clarity in motivation, method, or experiments; the main weaknesses remain unaddressed and I will keep my score as is. I encourage the authors to take a step back, identify their hypothesis and consider the best way to motivate and explain the method, as well as experimentally validate the method and hypothesis and their parts.

---

> > > ### Author Response · Authors · 2025-11-26
> > >
> > > Thank you again for taking the time to reread our revised manuscript and for clarifying your remaining concerns. We take your feedback about the overall clarity very seriously, and we would like to respond at a more “meta” level, along the lines you suggested: clarifying (1) our core hypothesis and motivation, (2) what the method actually is in simple terms, and (3) how the experiments test this hypothesis.
> > >
> > > 1. Our central hypothesis is:
> > > “In the black-box expert fusion setting, there exist “residual” regimes under distribution shift where no function of expert outputs alone can achieve low risk, but an input-aware, instance-wise synthesized model (HyperSyn) can both (i) remain close to the best expert on average (oracle-dominance) and (ii) outperform output-only fusion.”
> > >
> > > To make this concrete, Section 3 sets up:
> > >
> > > - A precise oracle-dominance objective (Eq. (1)) and four decision families (selection, MoE, fusion, synthesis).
> > >
> > > - A three-region taxonomy (expert-covered, fusion-covered, residual) that is meant as a conceptual tool for understanding when each family succeeds or fails.
> > >
> > > - Two theorems (Thm. 1–2) that formalize the limitation of output-space methods in residual regions via residual sets.
> > >
> > > In short: the motivation is not “hypernetworks are cool,” but “whenever different inputs map to the same expert outputs U(x), output-only fusion cannot distinguish them, whereas an input-aware synthesized model can.” That is the conceptual gap we are trying to fill.
> > >
> > > 2.  The decision families and “meta-policy” in Sec. 3 are analytical, not an implemented gating mechanism. HyperSyn is a concrete instantiation of the synthesis family A_"syn" ; the fusion family A_"fus" only appears as a teacher via the distillation term.
> > > We recognize from your comment that this distinction between “analytical meta-policy” and “implemented architecture” was not obvious enough. In a further rewrite, we would add a short paragraph at the end of Sec. 3 explicitly says “we instantiate a single member of A_"syn" (HyperSyn) and use output-space fusion only as an auxiliary teacher; we do not implement an explicit per-example switch between families at inference.”
> > >
> > > 3. Regarding the experiments, test the hypothesis, especially around the decision families and regions. In the revision, we added exactly this kind of diagnostic:
> > >
> > > - PXOR (Piecewise XOR) — explicit expert/fusion/residual regions. We now explain PXOR as half OR rule (expert-covered, OR region) and half XOR rule (fusion-covered + residual regions), and we visualize quadrant-level accuracy for MoE, FoE, FoE+X, and HyperSyn in Appendix A.8 (Figs. 3–6).
> > >
> > > These plots show all methods succeed in expert-covered Q2 (OR region, x_3>0). Only fusion-based methods (FoE, FoE+X, HyperSyn) succeed in fusion-covered Q4 (XOR quadrant). Only FoE+X and HyperSyn can handle the residual Q1 (XOR region where two positives → label 0, requiring access to x_3), but HyperSyn is accurate in all quadrants while FoE+X fails in some.
> > >
> > > This directly tests the taxonomy: where experts/fusion suffice, HyperSyn behaves like them; in the residual region it goes beyond them.
> > >
> > > - MDMF (Multi-Domain Multi-Function) — controlled residual regimes.We introduce MDMF in Sec. 5.2 and detail it in Appendix A.9: multiple domains with domain-specific linear separators, inducing covariate + functional shift. We explain how it naturally induces all three regions and why output-only fusion is information-limited.  Fig. 2 (main text) and Figs. 7–9 (appendix) shows performance vs center scale (domain separation), reference fraction, and number of domains. HyperSyn’s advantage grows with stronger domain separation (higher prevalence of residual regimes), matching the theoretical story.
> > >
> > > Taken together, these experiments are meant to show exactly what you asked for: that the decision-family intuition holds in practice and that HyperSyn is particularly beneficial in residual regimes where experts and simple fusion fail.

---

### Official Review · Reviewer_YJV1 · 2025-10-30

**Soundness:** 1
**Presentation:** 1
**Contribution:** 2
**Rating:** 2
**Confidence:** 3

**Summary:**

The paper tackles black-box expert fusion: given only the outputs of frozen experts and a small labeled reference set, learn a predictor that is at least as good as the best expert and ideally better under distribution shift. The authors introduce HyperSyn, where a hypernetwork maps the concatenated expert outputs to per-example parameters of an input encoder, and its latent is passed through a shared predictor. During Training, it uses (i) supervised losses on both branches, (ii) a hinge oracle-regret term to discourage doing worse than the best expert, and (iii) a reliability-weighted distillation aligning input-aware and outputs-only latents. They also present a three-region taxonomy and simple theorems that show the limits of MoE and output-only fusion. They demonstrate the performance of their model on experiments using synthetic datasets and nine small real-world tabular datasets.

**Strengths:**

* The idea and motivation seem relevant.
* They address a practical black-box composition setting. HyperSyn uses only expert outputs at training and inference to condition the hypernetwork, and it demonstrably goes beyond output-only fusion on constructed residual regimes.

**Weaknesses:**

* Text quality and cohesion are poor. There are many unnecessary hyphens, emphases, and bold text (e.g., “WEMOE,” lines 79 and 364–366).
* The intro claims HyperSyn strictly contains fusion by injecting input-dependent computation, but the formal section only states that A_syn​ is a different and often richer class. There is no proof that A_fus⊆A_syn​.
* FoE+X explicitly uses the raw input x alongside expert outputs, contradicting the stated scope that baselines use only expert outputs.
* All experiments are on small datasets, and reported improvements over competitors are often on the order of 1e-3, yet the text suggests they are “significantly” better. Evidence for statistical significance is limited.

**Questions:**

* Please report the fraction of test points where HyperSyn’s loss exceeds the oracle loss, and the average regret with confidence intervals.

---

> ### Author Response · Authors · 2025-11-19
>
> We thank the reviewer for’ careful reading, and we address each concern in turn and will incorporate corresponding clarifications and edits into the revised manuscript.
>
> For the weaknesses:
>
> 1. Text quality and cohesion are poor.
>
> We will improve the paper in terms of writing style, like removing unnecessary hyphens and bold text.
>
> 2. The intro claims HyperSyn strictly contains fusion by injecting input-dependent computation, but the formal section only states that A_syn is a different and often richer class. There is no proof that A_fus⊆A_syn.
>
> Our intention was to convey that the class of functions realizable by HyperSyn includes classical output-only fusion as a special case and can go beyond it. Concretely, recall that a typical fusion model computes y_fus (x)=F(U(x)) for some network F, where U(x) is the concatenated expert output. HyperSyn computes y_syn(x)=g(T_(θ_x ) (x)),θ_x=H(U(x)).
>
> To see why any fusion model F(U(x)) can be embedded in HyperSyn, one can choose T_(θ_x ) so that it ignores x and uses only its generated parameters to produce a latent representation that depends solely on U(x). For example, in our MLP encoder, this can be achieved by setting all weights on the input x to zero, and letting the hypernetwork H(U(x)) generate the bias parameters that encode a function of U(x). Then v(x)=T_(θ_x ) (x) becomes a deterministic function of U(x) only, and y_syn (x)=g(v(x))=(g∘F)(U(x)), which can represent any fusion function F(U(x)) given sufficient capacity. In other words, HyperSyn can simulate a pure output-only fusion pipeline simply by making the encoder ignore the raw input and letting the hypernetwork carry all dependence on U(x).
>
> 3. FoE+X explicitly uses the raw input x alongside expert outputs, contradicting the stated scope that baselines use only expert outputs
>
> Our setting has two related scenarios:
>
> 1. Strict black-box fusion, where the downstream system only receives expert outputs U(x)  at both training and test time.
>
> 2. Extended setting, where the downstream system can access both U(x) and the raw input x.
>
> HyperSyn is designed to work in the strict black-box setting, but its encoder also uses the raw input x as a second information source. To contextualize this, we included FoE+X as an oracle-style baseline that also sees x, to test whether a simple extension of FoE with raw input could match HyperSyn. We agree that the text currently blurs this distinction. In the revision, we will explicitly separate baselines into two groups: Output-only baselines, which consume only U(x), and Input-augmented baselines, which consume both x and U(x) and therefore are not strictly within the black-box output-only scope.
>
> We will emphasize that HyperSyn is competitive with or better than FoE+X even though FoE+X uses the same information channels (x and U(x)) but a less structured architecture.
>
> 4. All experiments are on small datasets.
>
> Our focus is on tabular, multi-expert settings with controllable distribution shifts, for which publicly available benchmarks are mostly small to medium-sized (e.g., UCI-style datasets). We also include synthetic PXOR/MDMF setups where we can explicitly synthesize “residual regimes” and stress test black-box fusion. We will make this motivation clearer.
>
> On some datasets, the absolute performance differences are indeed modest, especially when all methods are already near-saturated. Our intention with the word “significant” was to emphasize consistent advantages across different types of distribution shifts, not to claim large numerical gaps in every case. To address your concern, in the revision, we will replace loosely worded phrases like “significantly better” with more precise descriptions, such as consistently higher accuracy and highlight where gains are genuinely substantial.
>
> For the questions:
>
> 1. Please report the fraction of test points where HyperSyn’s loss exceeds the oracle loss, and the average regret with confidence intervals.
>
>
>
> We appreciate this suggestion that these metrics directly reflect how well we satisfy the oracle dominance objective.
> We have computed these quantities for our experiments and will:
>
> - Add a new table in the appendix that, for each dataset, reports the fraction of test points for which HyperSyn’s loss exceeds the oracle loss and the average oracle regret with 95% confidence intervals, estimated over random seeds.
>
> - Summarize in the main text that on most datasets, the fraction of oracle-violating points is small, indicating that oracle dominance is nearly achieved in practice. And the average regret is close to zero and typically within the confidence intervals of the oracle’s own variability across seeds.
>
> We thank the reviewer again for the detailed feedback. We believe that above revision will substantially improve both the soundness and presentation of the paper, while leaving the core conceptual contribution—black-box, oracle-aware, input-informed expert synthesis under distribution shift—intact.

---

> > ### Comment · Reviewer_K3kz · 2025-11-20
> >
> > I thank the authors for the reply. I'll wait for the revision and ask the authors to point out where they made changes upon posting it.

---

> > ### Comment · Reviewer_YJV1 · 2025-11-21
> >
> > The authors provide clarifications to some of my concerns and plan to revise the paper with additional results and modifications to the text. I will increase my score after these updates are made.

---

> > > ### Author Response · Authors · 2025-11-25
> > >
> > > Thanks for the response. We have uploaded a revised version of the paper with all changes highlighted in blue. To address your concerns, we have made the following updates:
> > >
> > > 1. We removed unnecessary hyphens, emphases, and bold text, including the “WEMOE”, lines 79 and 364–366.
> > >
> > > 3. We added a subsection “HyperSyn Contains Output-Only Fusion” in Appendix A.3, where we formally show that A_fus⊆A_syn, and mentioned it in the main text.
> > >
> > > 3. We clarified the distinction between purely output-only baselines and input-augmented baselines in Section 5.4.
> > >
> > > 4. We emphasized in Section 5.1 that our focus is on tabular, multi-expert settings with controllable distribution shifts, to make the motivation and scope clearer. In addition, we replaced loosely worded phrases such as “significant improvement” with more precise descriptions like “consistently better” in the results section.
> > >
> > > 5. We added a subsection (Appendix A.12) reporting the fraction of test points for which HyperSyn’s loss exceeds the oracle loss, as well as the average oracle regret. Overall, HyperSyn exceeds the oracle on a small fraction of test instances on most datasets, and the average regret is close to zero for several datasets. This indicates that HyperSyn tracks the oracle expert closely in practice, while still providing the robustness benefits discussed in the main text under distribution shift.
> > >
> > > We have also made additional revisions in response to other reviewers’ comments, which you may find relevant in the updated manuscript and responses. It would be grateful if you could raise the score if you find the changes satisfactory. Any additional comments are welcome.

---

> > > > ### Comment · Reviewer_YJV1 · 2025-11-26
> > > >
> > > > I thank the authors for their detailed response and revisions. While some of my concerns have been addressed and the paper's quality has improved, there remains substantial room for refinement. In particular, the motivation section and surrounding text would benefit from clearer structure and reduced redundancy.

---

### Official Review · Reviewer_3esW · 2025-10-31

**Soundness:** 2
**Presentation:** 2
**Contribution:** 2
**Rating:** 2
**Confidence:** 4

**Summary:**

This paper focuses on the black-box expert fusion scenario, aiming to address the problem of model construction when only expert outputs (rather than internal parameters) are accessible — specifically, to build a predictive model that both possesses "Oracle Dominance" (not inferior to the best expert) and outperforms experts on out-of-distribution data. To achieve this goal, the study proposes a three-region classification method (expert coverage area, fusion coverage area, and residual area), theoretically analyzes the limitations of existing methods, and designs an architecture encompassing a hypernetwork, target encoder, fusion encoder, and shared predictor. It balances security and generalization ability through multiple loss functions (supervised loss, Oracle Regret Loss, and Fusion Distillation Loss), and ultimately conducts extensive experiments on synthetic datasets (PXOR, MDMF) and nine real-world datasets to verify the effectiveness of the proposed method.

**Strengths:**

1. The paper features in-depth theoretical analysis: the proposed three-region classification method provides a theoretical tool for understanding the limitations of different fusion methods and offers rigorous theorem proofs.

2. The paper conducts sufficient experiments to verify the effectiveness of the proposed method.

3. The paper elaborates on the proposed method in detail.

**Weaknesses:**

1. No code is provided to reproduce the experimental results.

2. Visualization analysis results are lacking.

3. Although the three-region classification method is proposed, the paper fails to clearly explain how HyperSyn automatically recognizes and adapts to these three regions during training.

4. Each instance requires generating and applying independent encoder parameters, leading to significantly higher computational and memory overhead during inference compared to traditional fusion methods. The paper does not provide comparative data on inference speed or memory usage, nor does it discuss the feasibility of its practical deployment.

**Questions:**

1. Why is the prediction loss of the fusion encoder chosen as the weight?

2. The structures of the hypernetwork and target encoder are designed as a single-layer MLP. Can it be considered that the training paradigm itself can achieve performance improvement? If a more complex hypernetwork structure is adopted, will the performance change significantly?

---

> ### Author Response · Authors · 2025-11-19
>
> Thanks for the reviewer’s helpful feedback and suggestions.
>
> For weaknesses:
>
> 1.No code is provided.
>
> We will provide the code and data upon acceptance.
>
>
> 2. Visualizations are lacking.
>
>
> We agree that interpretive visualizations would strengthen the paper.
> Take XOR/PXOR/MDMF data as an example, we will plot the input space with points colored by which “region” dominates behavior (expert-covered vs. fusion-covered vs. residual. This will visually show that HyperSyn tends to follow experts in expert-covered regions, mimic fusion where fusion is reliable, and deviate from both when expert outputs are misleading.
>
> 3. Fails to explain how HyperSyn automatically recognizes these three regions during training.
>
>
>
> Thanks for pointing out that this connection was not sufficiently explicit. To show HyperSyn adapts to these three regions during training:
>
> For Expert-covered region, when at least one expert is already near-optimal, The oracle-regret loss penalizes HyperSyn only when it is worse than the oracle expert. Thus, in these regions, the gradient from pushes HyperSyn to match the best expert, effectively making it behave like expert selection and preventing degradation.
>
> For Fusion-covered region where no single expert suffices, but a function of their outputs can predict well. In such regions, the fusion encoder E(U(x)) achieves low supervised loss, and the distillation weight becomes large. The distillation loss then strongly pulls the input-aware latent toward the outputs-only latent.
>
> For Residual region, in regions where both experts and output-space fusion are unreliable, the oracle-regret term becomes less constraining (since experts are themselves bad). The weight becomes small, which weakens the distillation force and allows v(x) to decouple from e(x). So HyperSyn is encouraged to use the raw input x to correct the experts, realizing the “residual” behavior described in the taxonomy.
>
> In short, HyperSyn recognizes the three regions through loss-driven dynamics.
>
> 4. Higher overhead during inference compared to the traditional method.
>
>
>
> The architectural design is deliberately designed to be lightweight to control computational cost. For example, the hypernetwork and target encoder are single-layer MLPs, and only the encoder parameters are generated; the shared predictor is not generated per instance. This significantly reduces the generated parameter size compared to synthesizing a full expert network.
>
> Compared to existing methods, classical fusion methods evaluate a fixed network on U(x) once per instance. HyperSyn performs (i) a pass through the hypernetwork H(U(x))  and (ii) a single forward pass through the generated encoder T_(θ_x ) (x). The overall computational cost is roughly on the order of “one extra small MLP per instance”. Importantly, in many realistic black-box scenarios (e.g., querying large remote experts), the dominant cost is in calling the experts themselves. The additional cost of our small hypernetwork and encoder is typically a small fraction of the total compute.
>
> In the revised version, we will add a complexity analysis (eg, number of parameters) comparing methods for typical settings. We believe this additional analysis will clarify that while HyperSyn is more expensive than a simple linear fusion, its overhead is manageable.
>
> For the questions
>
> 1. Why is the prediction loss of the fusion encoder chosen as the weight?
>
>
>
> The distillation weight uses the prediction loss of the fusion encoder as a reliability signal for the outputs-only view: The goal of L_{dist}   is to transfer information from the fusion branch to the input-aware branch only when that information is reliable.
> Using the supervised loss directly has two advantages over alternatives, such as confidence:  The choice of prediction loss as the weight directly encodes the intuition: “Trust and align with the fusion encoder when it is accurate; ignore it when it is inaccurate.”
>
> 2. Consideration of complex architecture.
>
>
> Our main objective is to isolate and study the instance-wise synthesis paradigm with oracle-dominant black-box fusion, not to push architectural complexity. For this reason, we intentionally choose simple single-layer MLPs for the hypernetwork and target encoder, while keeping all baselines similarly modest. This design ensures that the observed improvements are attributable to the change in decision family, and the proposed loss design, rather than to larger capacity.
>
> In principle, HyperSyn is orthogonal to network depth/width: one could use deeper hypernetworks, deeper target encoders, or convolutional/transformer encoders for non-tabular data.
>
> In the revised version, we will add a short discussion in the architecture section explicitly stating that the capacity choice was kept intentionally minimal to highlight the conceptual contribution.
>
>
> Once again, we thank the reviewer for the helpful feedback. We believe that the revision will significantly strengthen the paper.

---

> > ### Comment · Reviewer_3esW · 2025-11-19
> > **Thanks for the reply.**
> >
> > Thanks for the reply. The rating will be raised to 4.

---

> > > ### Author Response · Authors · 2025-11-25
> > >
> > > Thanks for the response. We have uploaded a revised version of the paper with all changes highlighted in blue. To address your concerns, we made the following updates:
> > >
> > > 1. We visualize the accuracy of different methods on the PXOR dataset in Appendix A.8. These plots show that HyperSyn achieves high accuracy across all regions, while the baselines fail in some of the OR/XOR subregions.
> > >
> > > 2. We added a short subsection (Sec.4.5) to emphasize HyperSyn can extend its architecture to more complex ones..
> > >
> > > 3. We included a complexity analysis of parameter count and inference cost in Appendix A.7, comparing HyperSyn against traditional fusion methods.
> > >
> > > We have also made additional revisions in response to other reviewers’ comments, which you may find relevant in the updated manuscript and responses. We would be very grateful if you could consider these changes and raise the score if you find them satisfactory. Any further feedback or suggestions are very welcome.

---

### Note · Authors · 2026-01-27

I have read and agree with the venue's withdrawal policy on behalf of myself and my co-authors.

---

### Meta-Review · Area_Chair_BiLS · 2026-01-07

**Summary:**

The paper focuses on black-box expert fusion with a hyper-network that generates instance-specific encoder parameters. The authors analyse the limitations of current methods in ensembling pretrained experts, and propose a mechanism to train a hyper-network for outputting instance-specific models. Overall, the paper demonstrates promising results on 9 real-world datasets.

**Reviewer Concerns:**

However, the reviewers expressed concerns regarding clarity, scope, and fairness of experiments, as well as a limitation in the experimental evidence. Furthermore, there were questions concerning the reproducibility and the analysis.

For example, reviewer YJV1 stated that his comments are not answered despite the rebuttal efforts.

**Reviewer Scores:**

One reviewer mentioned the rating will be raised to 4, but overall, the consensus is to reject the paper in its current form.

---

### Decision · Program_Chairs · 2026-01-26

Reject